# Improving Reward Model Generalization from Adversarial Process Enhanced Preferences

Zhilong Zhang [1 2 3 *]  Tian Xu [1 2 3 *]  Xinghao Du [1 2 3 *]  Xingchen Cao [1 2 3]  Yihao Sun [1 2]  Yang Yu [1 2 3 †]

## Abstract

In sequential decision-making, the reward function serves as the primary supervision signal, guiding agents to acquire the desired behaviors. Traditional reward modeling methods rely heavily on human expertise, limiting their scalability. Automated preference generation from suboptimal demonstrations has emerged as a promising alternative to address this limitation. This approach first generates preference data from suboptimal demonstrations and then trains reward models based on these preferences. Despite its potential, existing methods often struggle to generate preference data with sufficient coverage, limiting the accuracy and generalizability of the resulting reward models. To overcome this limitation, we propose APEC (Automated Preference generation with Enhanced Coverage), a novel method that improves the coverage of preference data. APEC achieves this by selecting policy pairs with significantly different iteration indices from the whole adversarial imitation learning process. We provide a theoretical analysis to validate that the selected policy pairs provably hold preference relationships. Experimental results demonstrate that APEC consistently outperforms baseline methods in generating preferences with broader coverage across both vector-based and pixel-based control tasks. Consequently, the reward models trained with APEC align more closely with ground-truth rewards, deriving improved policy performance. Our code is released at https://github.com/Zzl35/APEC.

## 1. Introduction

In reinforcement learning (RL) (Sutton & Barto, 2018), the reward function serves as the cornerstone for defining the agent's objective, quantifying its performance, and steering it toward learning optimal behavior by maximizing cumulative rewards. However, designing an appropriate reward function remains a significant challenge. An ideal reward function should capture the intended behaviors while preventing the reward hacking issue that the policy produces artificially high rewards without achieving the desired behaviors (Gao et al., 2023). Moreover, it should provide dense and informative feedback, offering actionable guidance to boost efficient policy learning.

Traditional reward modeling methods can be broadly categorized into three types: manual design (Hwangbo et al., 2019; Kumar et al., 2021), inverse reinforcement learning (IRL) (Ng & Russell, 2000; Abbeel & Ng, 2004; Fu et al., 2017a; Luo et al., 2022), and preference-based reward learning (Bradley & Terry, 1952; Christiano et al., 2017; Wirth et al., 2017). Manual design involves crafting reward functions by hand, requiring extensive domain expertise and frequent adjustments. This approach is labor-intensive and often impractical for complex tasks. In contrast, IRL attempts to infer reward functions from expert demonstrations, eliminating the need for manual reward design. However, it suffers from inherent ambiguity and relies on optimal demonstrations (Amin & Singh, 2016; Kim et al., 2021), limiting its utility to tasks where such optimal demonstrations are difficult to obtain. Preference-based reward learning, which learns reward models from preference feedback using the Bradley-Terry model (Bradley & Terry, 1952), has emerged as a promising alternative. Despite this, existing methods typically require substantial human preference, limiting their applicability to large-scale tasks. In summary, traditional reward modeling methods rely heavily on human expertise, limiting their scalability.

To address these challenges, researchers have been exploring various solutions, with automated preference generation from suboptimal demonstrations emerging as a promising approach (Brown et al., 2020; Chen et al., 2021; Huo et al., 2023; Huang et al., 2023). For example, D-REX (Brown et al., 2020) starts by learning a behavior cloning (BC)

∗: Equal contribution. †: Corresponding author. [1]National Key Laboratory for Novel Software Technology, Nanjing University [2]School of Artificial Intelligence, Nanjing University, China [3]Polixir.ai. Correspondence to: Yang Yu <yuy@nju.edu.cn>.

*Proceedings of the $42^{nd}$ International Conference on Machine Learning*, Vancouver, Canada. PMLR 267, 2025. Copyright 2025 by the author(s).

(Pomerleau, 1991) policy from suboptimal demonstrations and then generates trajectories with varying preference levels by introducing different uniformly random noise to the BC policy. Based on this, LERP (Huo et al., 2023) improves the utility of the preference by adding noise directly into the reward space. Despite these innovations, such methods overlook the importance of the coverage of generated preferences, which is proven to be crucial for learning accurate reward models (Zhu et al., 2023; Nika et al., 2024). As a result, the reward models learned by these methods are not robust and thus cannot be used for subsequent policy learning. The purpose of this paper is to address the following research problem.

*How to automatically generate accurate preferences with broad coverage?*

In this paper, we present APEC, an **A**utomated **P**reference generation with **E**nhanced **C**overage method. This method is inspired by the key observation in adversarial imitation learning (AIL) (Ho & Ermon, 2016; Kostrikov et al., 2019; Garg et al., 2021; Xu et al., 2024), the value of the learned policy gradually increases as the training progresses, suggesting that later policies are preferred over earlier ones. Based on existing analysis on AIL (Xu et al., 2023), we provide a theoretical justification for this observation, demonstrating that the value gap bound of the learned policy diminishes as the iteration index increases. Building on this insight, APEC chooses policy pairs with significantly different iteration indexes from the whole AIL training process and then executes them to generate the preference data. In stark contrast to previous methods (Brown et al., 2020; Huo et al., 2023), which generate preference data by merely injecting uniformly random noise into a fixed policy, APEC leverages a wide range of policies from the whole AIL training process, resulting in a preference dataset with broad coverage.

Empirically, we evaluate APEC across five vector-based and three pixel-based continuous control tasks. The empirical results demonstrate that, compared to existing baselines, our method can generate preferences with broader coverage, and the reward models learned from these preferences align more closely with the true environment rewards. In particular, APEC is the only method that consistently produces policies that achieve performance comparable to or even better than demonstrations in all tasks.

We highlight the main contributions of our work below.

- We propose an automated preference generation method, APEC, which leverages the policy pairs with different iteration indexes from the AIL training process, ensuring broad coverage in the generated preference data.

- We provide a theoretical justification for APEC, demonstrating that the value gap bound of the learned policy diminishes as the iteration index increases.

- We validate APEC across eight continuous control tasks, demonstrating that it can learn reward models that align more closely with true rewards, leading to improved policy performance.

## 2. Preliminaries

**Markov Decision Process.** In this paper, we consider finite-horizon Markov Decision Processes (MDPs), represented by the tuple $\mathcal{M} = (\mathcal{S}, \mathcal{A}, P, r^{\star}, H, \rho)$. Here, $\mathcal{S}$ and $\mathcal{A}$ denote the state and action spaces, respectively. $H$ signifies the planning horizon, while $\rho$ stands for the initial state distribution. Besides, $P : \mathcal{S} \times \mathcal{A} \to \Delta(\mathcal{S})$ denotes the transition function where $P(s'|s, a)$ determines the probability of transiting to state $s'$ given state $s$ and action $a$. Similarly, $r^{\star} : \mathcal{S} \times \mathcal{A} \to [0, 1]$ outlines the reward function of this MDP. Without loss of generality, we assume that the reward value is bounded in $[0, 1]$. A non-stationary policy is denoted by $\pi = \{\pi_h\}_{h=1}^{H}$ with $\pi_h : \mathcal{S} \to \Delta(\mathcal{A})$, where $\Delta(\mathcal{A})$ denotes the probability simplex. Here, $\pi_h(a|s)$ represents the probability of selecting action $a$ in state $s$ at time step $h$, for $h \in [H]$. The quality of policy $\pi$ is evaluated by policy value: $V^{\pi} = \mathbb{E}\left[\sum_{h=1}^{H} r^{\star}(s_h, a_h); \pi, P\right]$. Besides, for a policy $\pi$, we define the state-action distribution $d^{\pi}(s, a) = (1/H) \cdot \sum_{h=1}^{H} \mathbb{P}(s_h = s, a_h = a; \pi)$.

**Reward Learning from Suboptimal Demonstrations.** This work studies the setup of reward learning from suboptimal demonstrations, which aims to infer reward functions from imperfect trajectories (Brown et al., 2019; 2020). In this setup, we typically assume the existence of a suboptimal policy, $\pi^{\mathrm{O}}$, that interacts with the environment to generate a suboptimal dataset. Formally, $\mathcal{D}^{\mathrm{O}} = \{\tau_n^{\mathrm{O}} = (s_1, a_1, s_2, a_2, \ldots, s_H, a_H); a_h \sim \pi_h^{\mathrm{O}}(\cdot|s_h), s_{h+1} \sim P(\cdot|s_h, a_h), \forall h \in [H]\}_{n=1}^{N}$. The learner then uses $\mathcal{D}^{\mathrm{O}}$ to infer a reward function $r_\phi$ parameterized by $\phi$, which is subsequently employed to derive an effective policy.

**Preference-based Reward Learning.** Preference-based reward learning (Christiano et al., 2017; Brown et al., 2019) aims to infer rewards from a preference dataset. These methods are typically built on the Bradley-Terry (BT) model (Bradley & Terry, 1952). In particular, for a pair of trajectories $(\tau^1, \tau^2)$, the BT model assumes that the probability of $\tau^1$ being preferred over $\tau^2$ can be formulated as

$$\mathbb{P}\left(\tau^1 \succ \tau^2\right) = \sigma(r^{\star}(\tau^1) - r^{\star}(\tau^2)).$$

Here $\sigma(x) = 1/(1 + \exp(-x))$ is the logistic function and $r^{\star}(\tau) = \sum_{h=1}^{H} r^{\star}(s_h, a_h)$ denotes the return of the trajectory $\tau = (s_1, a_1, \ldots, s_H, a_H)$. Based on this probability

modeling, these methods apply maximum likelihood estimation to learn rewards from the preference dataset $\mathcal{D}^{\mathrm{pref}}$.

$$\min_\phi -\mathbb{E}_{(\tau^+, \tau^-) \sim \mathcal{D}^{\mathrm{pref}}} \left[ \log \left( \sigma(r_\phi(\tau^+) - r_\phi(\tau^-)) \right) \right].$$

Here $(\tau^+, \tau^-)$ means that $\tau^+$ is preferred over $\tau^-$.

## 3. Method

In this section, we introduce APEC: Automated Preference Generation with Enhanced Coverage. We first highlight the importance of preference data coverage in preference-based reward learning, a critical issue that previous methods have largely overlooked (Section 3.1). Next, we present the key observation behind APEC: adversarial imitation learning (AIL) can produce a series of policies with increasing values. We also provide a theoretical justification for this observation (Section 3.2). Finally, we describe the resulting APEC method, which incorporates additional techniques such as the *Wasserstein distance criterion* and *random segment cropping* to further enhance the generalization of reward models (Section 3.3).

### 3.1. The Coverage of Preference Data is Crucial

This work follows the pipeline of first automatically generating a preference dataset and then learning rewards from it. In this preference-based reward learning procedure, the coverage of the preference dataset is a pivotal factor (Zhu et al., 2023; Nika et al., 2024). Specifically, (Zhu et al., 2023, Theorem 5.2) indicates that if policy optimization is performed using the reward learned from the preference dataset, the resulting policy exhibits a sub-optimality bound that depends on the following coverage coefficient.

$$\mathrm{Cov}(\mathcal{D}^{\mathrm{pref}}) = \left\| (\Sigma_{\mathcal{D}^{\mathrm{pref}}} + \lambda I)^{-1/2} \mathbb{E}_{(s,a) \sim d^{\pi^\star}} [\phi(s,a)] \right\|_2.$$

Here $\phi : \mathcal{S} \times \mathcal{A} \to \mathbb{R}^d$ is the feature function. For a trajectory $\tau$, we use $\phi(\tau) = \sum_{(s,a) \in \tau} \phi(s,a)$ to denote the summation of features along this trajectory and $\Sigma_{\mathcal{D}^{\mathrm{pref}}} = (1/|\mathcal{D}^{\mathrm{pref}}|) \cdot \sum_{(\tau^+, \tau^-) \in \mathcal{D}^{\mathrm{pref}}} (\phi(\tau^+) - \phi(\tau^-))(\phi(\tau^+) - \phi(\tau^-))^\top \in \mathbb{R}^{d \times d}$ is the covariance matrix of $\mathcal{D}^{\mathrm{pref}}$. $\mathrm{Cov}(\mathcal{D}^{\mathrm{pref}})$ qualifies the coverage of $\mathcal{D}^{\mathrm{pref}}$ over the target data distribution induced by the optimal policy $\pi^\star$ in the feature space. If $\mathcal{D}^{\mathrm{pref}}$ exhibits good coverage in the sense that $\mathrm{Cov}(\mathcal{D}^{\mathrm{pref}})$ is small, the derived policy enjoys a small sub-optimality.

However, previous automated preference generation methods (Brown et al., 2020; Huo et al., 2023) often overlook the crucial role of coverage and struggle to generate a preference dataset with sufficient coverage. For instance, D-REX (Brown et al., 2020) first learns a behavioral cloning policy $\pi^{\mathrm{BC}}$ from $\mathcal{D}^{\mathrm{O}}$. D-REX then designs a disturbed policy $\pi_h^\varepsilon(\cdot|s) = (1 - \varepsilon)\pi_h^{\mathrm{BC}}(\cdot|s) + \varepsilon \, \mathrm{Unif}_{\mathcal{A}}(\cdot)$, where $\varepsilon$ controls

the level of disturbance, and $\mathrm{Unif}_{\mathcal{A}}(\cdot)$ denotes the uniform distribution over the action space. Finally, D-REX generates preferences by executing these disturbed policies with different values of $\varepsilon$, i.e., $(\tau^+, \tau^-)$ if $\tau^+ \sim \pi^\varepsilon$ and $\tau^- \sim \pi^{\varepsilon'}$ with $\varepsilon < \varepsilon'$. Consequently, the resulting preference data distribution is induced by these disturbed policies. We argue that this distribution exhibits poor coverage. In particular, due to the well-known compounding errors issue (Syed & Schapire, 2010), the BC policy $\pi^{\mathrm{BC}}$ suffers a significant performance gap compared to $\pi^{\mathrm{O}}$. As a result, these disturbed policies, interpolated between $\pi^{\mathrm{BC}}$ and $\mathrm{Unif}_{\mathcal{A}}$, fail to provide adequate coverage of high-return regions. We also empirically validate this claim. As shown in Figure 4, the preference data distributions generated by D-REX exhibit relatively poor coverage.

### 3.2. AIL Produces Policy Pairs with Clear Preferences and Good Coverage

This work aims to enhance previous automatic preference generation methods by improving preference data coverage. The proposed method is inspired by the following key observation: in another imitation learning method, adversarial imitation learning (AIL), the value of the learned policy gradually increases as training progresses, as shown in Figure 2. We briefly introduce AIL in the following. AIL mimics $\mathcal{D}^{\mathrm{O}}$ by solving the minimax optimization problem of:

$$\min_\pi \max_r \mathbb{E}_{\tau \sim \mathcal{D}^{\mathrm{O}}} \left[ \sum_{h=1}^H r(s_h, a_h) \right] - \mathbb{E}_{\tau \sim \pi} \left[ \sum_{h=1}^H r(s_h, a_h) \right].$$

For the inner optimization, the variable $r : \mathcal{S} \times \mathcal{A} \to [0,1]$ can be regarded as a reward function, aiming to maximize the value gap between $\pi^{\mathrm{O}}$ and $\pi$. To solve the above minimax problem, AIL methods alternatively update the policy variable and reward variable in an iterated manner; see Algorithm 1 for a standard AIL training procedure. Proposition 3.1 provides a theoretical justification for the above observation on the AIL training process.

**Proposition 3.1.** *Consider the AIL method shown in Algorithm 1. For any fixed $\delta \in (0,1]$, with probability at least $1 - \delta$, we have*

$$V^{\pi^{\mathrm{O}}} - V^{\pi^k} \leq 4H^2 \sqrt{\frac{2|\mathcal{S}||\mathcal{A}| \ln(|\mathcal{A}|)}{k}} + \varepsilon_{\mathrm{stat}}.$$

*Here $\pi^k$ denotes the policy in the k-th iteration and $\varepsilon_{\mathrm{stat}} = 4\sqrt{2}H(|\mathcal{S}||\mathcal{A}| \log(1/\delta))^{\frac{1}{2}}|\mathcal{D}^{\mathrm{O}}|^{-\frac{1}{2}}$ denotes the statistical error due to finite samples in $\mathcal{D}^{\mathrm{O}}$.*

The proof is deferred to Appendix A. Proposition 3.1 demonstrates that during the AIL training process, the value gap bound of the learned policy diminishes as the iteration index

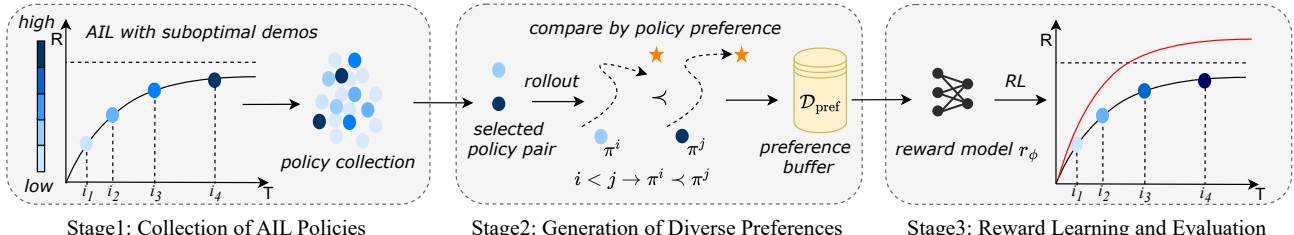

Stage1: Collection of AIL Policies    Stage2: Generation of Diverse Preferences    Stage3: Reward Learning and Evaluation

*Figure 1.* Overview of APEC. There are three key stages: (1) Collecting policies from the AIL training process. (2) Generating diverse preferences based on their corresponding iteration indexes. (3) Learning generalizable reward models from generated preferences.

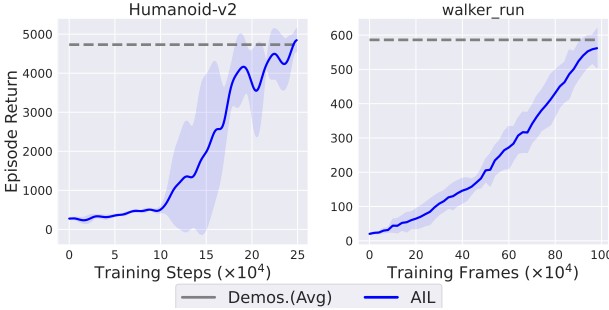

*Figure 2.* The learning curves of AIL on Mujoco and DMControl tasks. The x-axis represents the number of interactions with the environment (frames for pixel-based tasks), and the y-axis represents the policy return. The gray dashed line represents the average return of trajectories in suboptimal demonstrations $\mathcal{D}^O$.

increases. This may indicate a progressive improvement in policy value over successive iterations, suggesting that later policies are preferred over earlier ones.

Building on this insight, we propose constructing policy pairs with distinct preferences by repeatedly selecting two policies with significantly different iteration indices from the AIL training process. These policy pairs are then used to generate the preference dataset. In our approach, the preference data distribution is induced by policies across the entire AIL training process, ensuring broad coverage for two reasons. First, both theoretical (Xu et al., 2020; Rajaraman et al., 2020) and practical (Ho & Ermon, 2016; Kostrikov et al., 2019) works have demonstrated that AIL effectively mitigates the compounding errors issue in BC, leading to final policies that closely match $\pi^O$ and induce a distribution with good coverage of high-return regions. Second, by drawing from policies throughout the entire AIL training process, the preference dataset captures a broad and representative coverage of the whole state-action space.

### 3.3. APEC: Automated Preference Generation with Enhanced Coverage

We present our method, APEC, which consists of three components: collection of AIL policies, generation of di-

verse preferences, and reward learning and evaluation, as illustrated in Figure 1.

**Collection of AIL Policies.** In Section 3.2, we prove that the policy performance bound of the AIL algorithm improves for training epochs. For our implementation, we use DAC (Kostrikov et al., 2019), a well-known AIL algorithm recognized for its excellent sample efficiency and convergence properties. Additionally, we save checkpoints of the policy model at regular intervals during training. While these policies display diverse behaviors, they consistently show a steady increase in expected cumulative return, which is then used to generate preferences for subsequent steps.

**Generation of Diverse Preferences.** After policy collection, we construct a preference buffer. Following the guidance of Proposition 3.1, we iteratively sample policy pairs $\pi_i$ and $\pi_j$ from the collected policy set, ensuring that $j - i > k$, where $k$ is the training epoch interval between two policies. For each pair, we allow $\pi_i$ and $\pi_j$ to rollout trajectories $\tau_i$ and $\tau_j$ in the environment, respectively. However, directly adding $(\tau^i, \tau^j, \succ)$ to the preference buffer presents two challenges: 1) In tasks with high stochasticity, the performance variance is high among trajectories sampled from the same policy. Although $V(\pi^j)$ is expected to be greater than $V(\pi^i)$, the instance-level performance $R(\tau^i) < R(\tau^j)$ may not always hold. 2) During training, AIL may experience occasional performance spikes, where policy performance drops dramatically but recovers quickly. To address the above two issues, we introduce a Wasserstein distance criterion for doubly robust preference generation. Specifically, we calculate the Wasserstein distance between sampled trajectories and suboptimal demonstrations $\mathcal{D}^O$, denoted as $d_i = \frac{1}{N} \sum_{n=1}^{N} \mathcal{W}(\tau^i, \tau_n^O), d_j = \frac{1}{N} \sum_{n=1}^{N} \mathcal{W}(\tau^j, \tau_n^O)$. We then impose the constraint $d_i - d_j > \delta$ for a threshold $\delta$. Finally, only trajectory pairs that satisfy both the epoch criterion and the Wasserstein distance criterion conditions are added to the preference buffer. In Section 4.5, we validate the effectiveness of the Wasserstein distance criterion through ablation experiments.

**Reward Learning.** Reward learning from preferences aims at estimating the reward model that reflects the observed

preferences. Most approaches are based on the Bradley-Terry model (Bradley & Terry, 1952), which assumes that the probability of preferring one trajectory segment over another is exponentially related to the sum of the reward model $r_\phi$ estimates for those segments. However, we have found that when the trajectory horizon is long and preference data consist of full trajectories, directly applying the Bradley-Terry model often fails to capture the desired behavior in complex tasks. To address this, we randomly extract segments from full trajectories and use these partial trajectories for training as proposed in T-REX (Brown et al., 2019). The loss function for this approach is defined in Equation (1).

$$\min_\phi -\mathbb{E}_{\substack{h \sim [0, H-l], \\ (\tau^-, \tau^+) \sim \mathcal{D}}} \left[ \log \left( \sigma(r_\phi(\tau^+_{[h:h+l]}) - r_\phi(\tau^-_{[h:h+l]})) \right) \right]. \tag{1}$$

## 4. Experiment

We perform comprehensive experiments to answer the following questions. **Q1**: How accurate is the learned reward model? How does its performance compare to the baselines? (Section 4.2) **Q2**: Can the learned reward model derive an effective policy? (Section 4.3) **Q3**: How accurate are the generated preferences? Do they offer broader coverage than baselines? (Section 4.4) **Q4**: How does each component in APEC affect performance? (Section 4.5)

### 4.1. Experimental Setup

**Benchmark.** We evaluate our method on five tasks from the feature-based Mujoco benchmark (Todorov et al., 2012), and three tasks from the pixel-based DMControl benchmark (Tassa et al., 2018), which are leading benchmarks in reinforcement learning and imitation learning that provide a diverse set of continuous control tasks.

**Suboptimal demonstrations and test dataset.** We train RL agents using SAC (Haarnoja et al., 2018) for MuJoCo tasks and DrQ-v2 (Yarats et al.) for DMControl tasks, ensuring sufficient training epochs and treating the final policy checkpoint as the optimal one. To generate suboptimal demonstrations, we select policy checkpoints with performance ranging from 50% to 80% of the optimal. Additionally, we uniformly sample 1,000 trajectories from the replay buffers during training to create test datasets, enabling the evaluation of the generalization of the learned reward functions across different algorithms.

**Primary Baselines.** We compare APEC with four state-of-the-art baselines: D-REX (Brown et al., 2020), AIRL (Fu et al., 2017a), SSRR (Chen et al., 2021), and LERP (Huo et al., 2023). To control for the impact of sample size on training results, we ensure that all methods are trained with an identical number of sampled trajectories. Specifically, we use 2000 trajectories for Mujoco and 10,000 trajectories

for DMControl.

**Experimental Challenges.** Compared to previous works, our experimental setup is significantly more challenging. (1) Pixel-based continuous control tasks are included. (2) Fewer demonstrations are available, for Mujoco tasks, only one suboptimal demonstration is provided, whereas for DMControl tasks, ten suboptimal demonstrations are used. (3) The suboptimal demonstrations are selected from a high-return range, making it more difficult to achieve better-than-demo performance.

### 4.2. Reward Model Accuracy

To address **Q1**, we compare the learned reward accuracy of APEC with that of other baselines. Specifically, we evaluate performance on the test dataset using three key metrics: (1) **Reward correlation**, the Pearson correlation between learned and true rewards. (2) **Return correlation**, the Pearson correlation between learned and true returns. (3) **Preference accuracy**, the accuracy of preference labels given by learned returns on randomly constructed trajectory pairs from the test dataset.

Tables 1 and 2 present the reward and return correlation results across various tasks and methods, highlighting the effectiveness of APEC compared to baseline approaches. Specifically, Table 1 shows that APEC achieves the highest reward correlation scores, consistently surpassing baselines. Notably, in the Ant and HalfCheetah tasks, APEC reaches reward correlation values of 0.94 and 0.91, respectively. Table 2 presents return correlation results, further reinforcing APEC's superiority in deriving high-quality policies. APEC consistently achieves the highest return correlation scores, reaching near-perfect alignment (0.99–1.00) in several vector-based tasks. The method also performs exceptionally well in pixel-based control environments, with return correlation values averaging 0.96, significantly higher than competing approaches. Overall, our method surpasses the baselines in nearly all tasks, indicating that the preferences generated by APEC lead to more accurate reward models.

| Environment | SSRR | AIRL | D-REX | LERP | APEC(ours) |
|---|---|---|---|---|---|
| Ant | $0.16_{\pm 0.17}$ | $0.61_{\pm 0.01}$ | $-0.07_{\pm 0.04}$ | $-0.09_{\pm 0.04}$ | $\mathbf{0.94}_{\pm 0.02}$ |
| HalfCheetah | $0.33_{\pm 0.12}$ | $0.48_{\pm 0.14}$ | $0.12_{\pm 0.09}$ | $0.12_{\pm 0.10}$ | $\mathbf{0.91}_{\pm 0.02}$ |
| Hopper | $0.37_{\pm 0.13}$ | $0.77_{\pm 0.05}$ | $0.00_{\pm 0.24}$ | $0.00_{\pm 0.24}$ | $\mathbf{0.83}_{\pm 0.05}$ |
| Humanoid | $0.05_{\pm 0.02}$ | $\mathbf{0.23}_{\pm 0.03}$ | $0.02_{\pm 0.01}$ | $0.02_{\pm 0.01}$ | $0.04_{\pm 0.04}$ |
| Walker2d | $0.25_{\pm 0.06}$ | $0.24_{\pm 0.07}$ | $0.19_{\pm 0.05}$ | $0.19_{\pm 0.07}$ | $\mathbf{0.88}_{\pm 0.02}$ |
| Average | 0.23 | 0.47 | 0.05 | 0.05 | **0.72** |
| cheetah_run | $-0.12_{\pm 0.02}$ | $-0.19_{\pm 0.03}$ | $0.68_{\pm 0.00}$ | $0.70_{\pm 0.00}$ | $\mathbf{0.73}_{\pm 0.00}$ |
| walker_run | $0.06_{\pm 0.18}$ | $-0.01_{\pm 0.24}$ | $0.60_{\pm 0.01}$ | $0.62_{\pm 0.00}$ | $\mathbf{0.88}_{\pm 0.00}$ |
| walker_walk | $0.06_{\pm 0.09}$ | $-0.17_{\pm 0.07}$ | $0.51_{\pm 0.02}$ | $0.54_{\pm 0.01}$ | $\mathbf{0.76}_{\pm 0.00}$ |
| Average | 0.00 | -0.12 | 0.59 | 0.62 | **0.79** |

*Table 1.* Reward correlation on Mujoco and DMControl tasks over 5 random seeds.

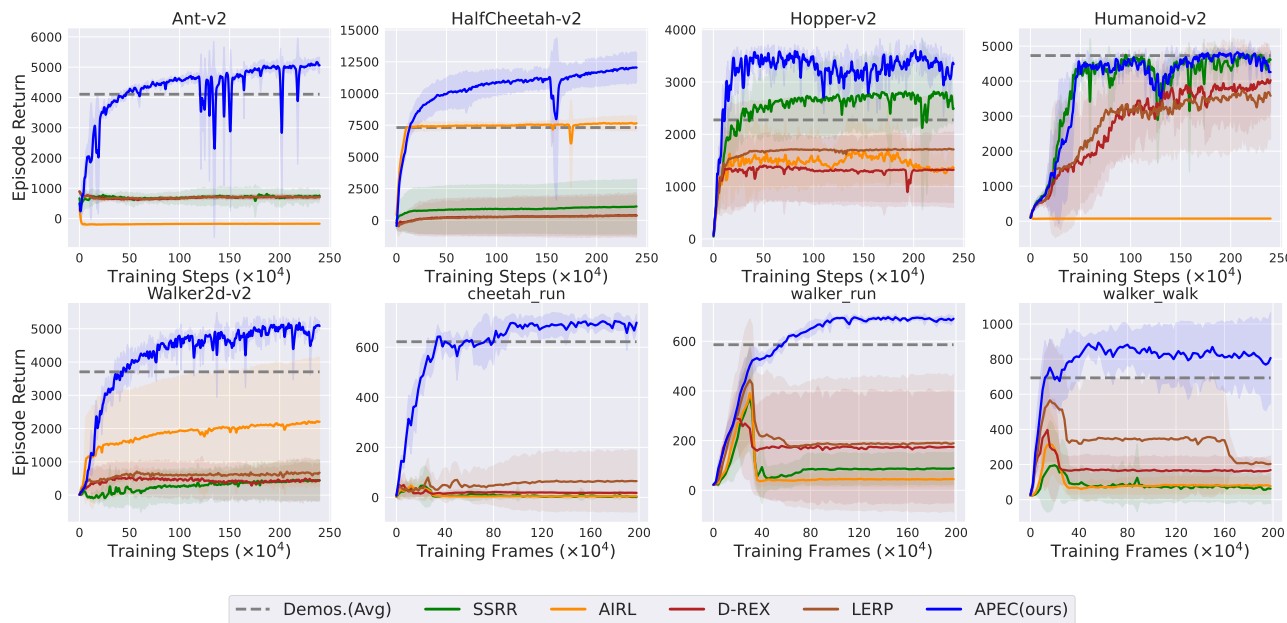

*Figure 3.* Overall performance on policy learning evaluation on Mujoco and DMControl tasks. Here the x-axis is the number of training steps (frames for pixel-based tasks) and the y-axis is the policy return. The solid lines are the mean of results while the shaded region corresponds to the standard deviation over 5 random seeds.

| Environment | SSRR | AIRL | D-REX | LERP | APEC(ours) |
|---|---|---|---|---|---|
| Ant | $0.57_{\pm 0.28}$ | $0.45_{\pm 0.01}$ | $-0.49_{\pm 0.35}$ | $-0.57_{\pm 0.28}$ | $\mathbf{0.99}_{\pm 0.00}$ |
| HalfCheetah | $0.92_{\pm 0.08}$ | $0.96_{\pm 0.02}$ | $0.42_{\pm 0.40}$ | $0.40_{\pm 0.44}$ | $\mathbf{0.99}_{\pm 0.00}$ |
| Hopper | $0.99_{\pm 0.00}$ | $0.97_{\pm 0.01}$ | $0.47_{\pm 0.71}$ | $0.46_{\pm 0.70}$ | $\mathbf{1.00}_{\pm 0.00}$ |
| Humanoid | $\mathbf{0.99}_{\pm 0.00}$ | $-0.99_{\pm 0.00}$ | $-0.87_{\pm 0.17}$ | $-0.89_{\pm 0.14}$ | $0.95_{\pm 0.03}$ |
| Walker2d | $0.98_{\pm 0.01}$ | $0.69_{\pm 0.18}$ | $0.75_{\pm 0.35}$ | $0.84_{\pm 0.22}$ | $\mathbf{1.00}_{\pm 0.00}$ |
| Average | 0.89 | 0.42 | 0.06 | 0.05 | **0.99** |
| cheetah_run | $-0.44_{\pm 0.21}$ | $-0.43_{\pm 0.37}$ | $0.94_{\pm 0.00}$ | $0.94_{\pm 0.00}$ | $\mathbf{0.98}_{\pm 0.00}$ |
| walker_run | $0.15_{\pm 0.30}$ | $-0.01_{\pm 0.56}$ | $0.86_{\pm 0.01}$ | $0.87_{\pm 0.01}$ | $\mathbf{0.98}_{\pm 0.00}$ |
| walker_walk | $0.12_{\pm 0.22}$ | $-0.23_{\pm 0.20}$ | $0.65_{\pm 0.02}$ | $0.70_{\pm 0.01}$ | $\mathbf{0.92}_{\pm 0.00}$ |
| Average | $-0.06$ | $-0.22$ | 0.81 | 0.84 | **0.96** |

*Table 2.* Return correlation on Mujoco and DMControl tasks over 5 random seeds.

| Environment | SSRR | AIRL | D-REX | LERP | APEC(ours) |
|---|---|---|---|---|---|
| Ant | $0.66_{\pm 0.11}$ | $0.69_{\pm 0.02}$ | $0.37_{\pm 0.08}$ | $0.34_{\pm 0.07}$ | $\mathbf{0.85}_{\pm 0.02}$ |
| HalfCheetah | $0.83_{\pm 0.11}$ | $0.90_{\pm 0.01}$ | $0.69_{\pm 0.10}$ | $0.69_{\pm 0.11}$ | $\mathbf{0.97}_{\pm 0.01}$ |
| Hopper | $0.91_{\pm 0.02}$ | $0.84_{\pm 0.01}$ | $0.68_{\pm 0.27}$ | $0.68_{\pm 0.26}$ | $\mathbf{0.95}_{\pm 0.00}$ |
| Humanoid | $\mathbf{0.88}_{\pm 0.01}$ | $0.10_{\pm 0.00}$ | $0.23_{\pm 0.10}$ | $0.23_{\pm 0.09}$ | $0.78_{\pm 0.06}$ |
| Walker2d | $0.88_{\pm 0.04}$ | $0.65_{\pm 0.05}$ | $0.80_{\pm 0.14}$ | $0.81_{\pm 0.13}$ | $\mathbf{0.95}_{\pm 0.01}$ |
| Average | 0.83 | 0.63 | 0.55 | 0.55 | **0.90** |
| cheetah_run | $0.39_{\pm 0.02}$ | $0.44_{\pm 0.26}$ | $0.87_{\pm 0.00}$ | $0.87_{\pm 0.00}$ | $\mathbf{0.92}_{\pm 0.00}$ |
| walker_run | $0.55_{\pm 0.01}$ | $0.55_{\pm 0.26}$ | $0.78_{\pm 0.01}$ | $0.78_{\pm 0.00}$ | $\mathbf{0.90}_{\pm 0.00}$ |
| walker_walk | $0.54_{\pm 0.01}$ | $0.41_{\pm 0.12}$ | $0.65_{\pm 0.00}$ | $0.65_{\pm 0.00}$ | $\mathbf{0.76}_{\pm 0.00}$ |
| Average | 0.49 | 0.47 | 0.77 | 0.77 | **0.86** |

*Table 3.* Preference accuracy on Mujoco and DMControl tasks over 5 random seeds.

Furthermore, our method outperforms others in labeling preferences for unseen trajectory pairs, as shown in Table 3. APEC achieves an average labeling accuracy of **90%** for Mujoco tasks and **86%** for DMControl tasks on the test dataset. In contrast, all other baselines fail to achieve 80% accuracy in any task category.

### 4.3. Policy Learning Evaulation

To answer **Q2**, we directly use the learned reward model for RL training and assess its reusability by comparing convergence performance. Specifically, we use SAC for Mujoco tasks and DrQ-v2 for DMControl tasks. Additionally, to accelerate convergence, the training buffer was initialized with trajectories sampled from the AIL step (Stage 1 in Figure 1). For Mujoco tasks, we uniformly sample 10

trajectories, while for DMControl tasks, all available trajectories are used. We believe that a well-trained reward model should enable the agent to achieve performance superior to suboptimal demonstrations.

The results of the policy learning evaluation are presented in Figure 3. APEC achieves better-than-demo performance on 7 out of the 8 tasks, and for the `Humanoid-v2`, it also demonstrates comparable performance, showcasing the strong generalization of our reward models. While other baselines have demonstrated effectiveness in prior works (Fu et al., 2017a; Chen et al., 2021; Huo et al., 2023), they struggle to match demonstration performance under our more challenging experimental setup, which involves fewer demonstrations, more complex tasks, and a stricter better-than-demo requirement. Our analysis reveals that

this is primarily caused by reward hacking in the baselines, whereas our method benefits from the broader coverage of generated data, thus avoiding this problem. Appendix C.2 provides a detailed discussion of reward hacking.

### 4.4. Generated Preference Analysis

To answer **Q3**, we analyze the generated preferences of D-REX, SSRR, and APEC from *generation accuracy* and *generation coverage* two perspectives.

**Generation accuracy.** The experimental results for generation accuracy are presented in Table 10. In both vector-based and pixel-based control tasks, APEC achieves an impressive accuracy of 98%, consistently outperforming the baseline methods by a significant margin. SSRR and D-REX, which are based on noise injection, achieve 92% and 80% respectively. This improvement in accuracy is critical for learning robust reward models, as it ensures that the preferences used for training are reliable.

| Environment | SSRR | D-REX | APEC(ours) |
|---|---|---|---|
| Ant | $0.11_{\pm0.00}$ | $0.19_{\pm0.00}$ | $\mathbf{0.09}_{\pm0.00}$ |
| HalfCheetah | $1.54_{\pm0.02}$ | $2.13_{\pm0.02}$ | $\mathbf{1.20}_{\pm0.05}$ |
| Hopper | $0.29_{\pm0.01}$ | $0.71_{\pm0.01}$ | $\mathbf{0.23}_{\pm0.00}$ |
| Humanoid | $\mathbf{0.96}_{\pm0.02}$ | $1.14_{\pm0.01}$ | $1.02_{\pm0.04}$ |
| Walker2d | $0.48_{\pm0.01}$ | $0.77_{\pm0.02}$ | $\mathbf{0.45}_{\pm0.00}$ |
| Average | 0.68 | 0.99 | **0.60** |
| cheetah_run | $0.90_{\pm0.00}$ | $0.85_{\pm0.00}$ | $\mathbf{0.82}_{\pm0.00}$ |
| walker_run | $0.68_{\pm0.00}$ | $0.79_{\pm0.00}$ | $\mathbf{0.63}_{\pm0.00}$ |
| walker_walk | $0.53_{\pm0.00}$ | $0.56_{\pm0.00}$ | $\mathbf{0.49}_{\pm0.00}$ |
| Average | 0.70 | 0.73 | **0.65** |

*Table 4.* Projection distances on Mujoco and DMControl tasks over 5 random seeds.

**Generation coverage.** To assess the coverage of the generated dataset, we first calculate the projection distance $d_{\mathrm{proj}}$ from the test dataset to the generated dataset as defined in Equation 2, where $\mathcal{D}^O$ and $\mathcal{D}^T$ are the generated dataset and test dataset respectively. A smaller projection distance typically indicates better sample coverage. The results presented in Table 4 show that APEC exhibits the smallest projection distance from the test dataset to the generated samples except for the humanoid task. We believe that APEC outperforms other baselines due to its ability to generate the most well-covered samples while maintaining high production accuracy. Additionally, we visualize the sample distributions for `Hopper-v2` and `walker_run` in Figure 4. These visualizations reveal that APEC-generated samples provide broader coverage in both the x-axis displacement and x-axis velocity dimensions, especially in high-return regions (the return of these locomotion tasks is generally strongly correlated with the x-axis displacement),

which is crucial for achieving better-than-demo policies.

$$d_{\mathrm{proj}} = \frac{1}{|\mathcal{D}^T|} \sum_{s \in \mathcal{D}^T} \min_{s' \in \mathcal{D}^O} \|s - s'\|_1 \qquad (2)$$

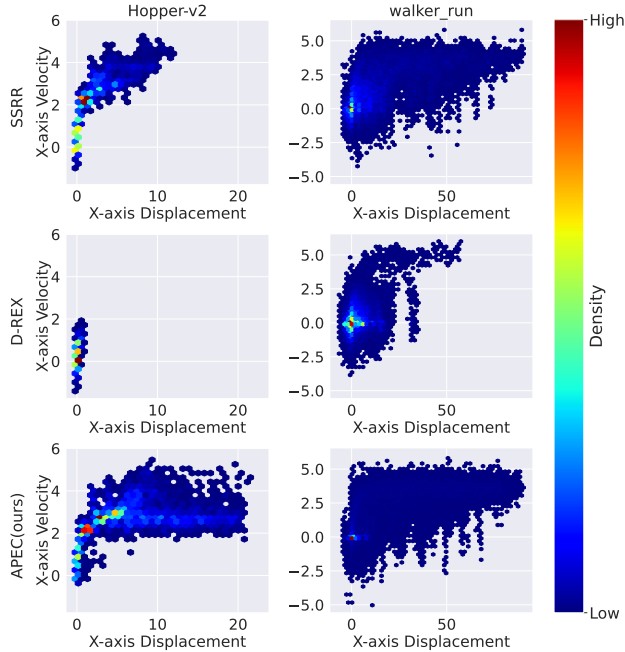

*Figure 4.* Visualization of generated preference distributions on Hopper-v2 and walker_run tasks. From top to bottom, the rows represent the generated samples for SSRR, D-REX, and APEC, respectively. In each plot, the x-axis represents the x-axis displacement, while the y-axis represents the x-axis velocity.

### 4.5. Ablation Study

In this section, we investigate the effect of the Wasserstein distance criterion and the number of collection policies on the generalization of the reward model. We use reward correlation mentioned in Section 4.2 as the evaluation metric. The results are shown in Figure 5.

**Wasserstein distance criterion.** We observe that the Wasserstein distance criterion is crucial to improving the accuracy of generated preferences. Its importance is especially evident in Mujoco tasks, where the training process may exhibit instability. By combining the epoch criterion with the Wasserstein distance criterion, we substantially enhance the generalization of reward models.

**Number of collection policies.** For the Mujoco tasks, we reduce the number of policies used for preference collection from 400 to 4, and for the DMControl tasks, from 50 to 5. Despite this reduction, the number of generated preferences remains constant (1000 for Mujoco and 5000 for DMControl). Our results reveal that using a smaller set of

policies decreases the diversity and coverage of the generated preferences, which in turn hampers the generalization of the reward function. In DMControl tasks, where the model inputs are images and the tasks are more complex, the diversity of the policies becomes even more critical for the generalization of reward models.

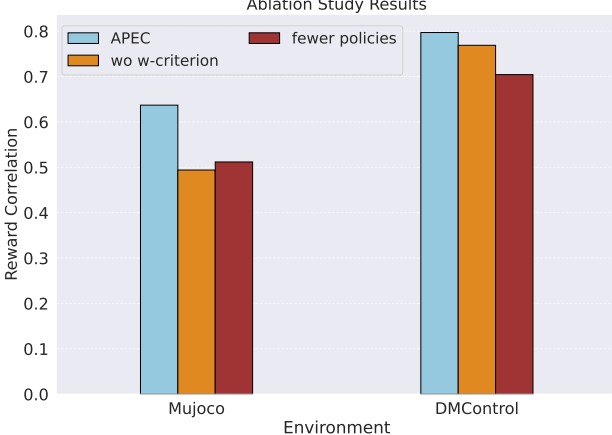

*Figure 5.* Ablation study on Mujoco and DMControl Tasks. We use reward correlation as the evaluation metric, with results averaged across all tasks over 5 random seeds. The blue bars represent APEC, the orange bars show the results without the Wasserstein distance criterion, and the red bars indicate the results obtained using fewer policies.

## 5. Related Works

**Reward Modeling.** Reward modeling is a critical aspect of reinforcement learning, traditionally relying on expert knowledge, which often requires significant human input and incurs considerable costs (Hadfield-Menell et al., 2017). One of the most widely used methods for reward modeling, inverse reinforcement learning (IRL), aims to recover reward functions from expert behaviors, thus minimizing human involvement in the reward design process. In apprenticeship learning (Abbeel & Ng, 2004), the reward model is optimized to maximize the performance gap between expert demonstrations and the learned policy. MaxEnt IRL (Ziebart et al., 2008) formulates reward updates as maximum likelihood estimation problems, integrating a maximum entropy principle to promote policy diversity and better represent expert behavior. As an extension of IRL, adversarial imitation learning (AIL) algorithms learn rewards through a fully adversarial process inspired by GANs (Goodfellow et al., 2014), achieving high data efficiency and replicating expert policies with minimal expert demonstrations. For instance, AIRL (Fu et al., 2017b) learns a reward function by training a discriminator to distinguish between expert and generated trajectories, resulting in a reward model that is robust to changes in dynamics and environmental properties. Un-

like IRL and AIL, preference-based reinforcement learning (PBRL) leverages human preferences to train reward models. The pioneering work by (Christiano et al., 2017) introduced neural networks to PBRL, demonstrating their capacity to capture complex behaviors. Building on this foundation, (Ibarz et al., 2018) improves the sample efficiency of PBRL by utilizing expert demonstrations, while (Cao et al., 2024) advances the approach by relaxing the optimality constraint on trajectories. Additionally, FTB (Zhang et al., 2023) uses a diffusion model for trajectory generation, achieving more robust policy learning by preferred trajectory augmentation. In contrast, APEC does not require human annotations or optimal trajectories and can achieve robust reward learning from a small number of suboptimal trajectories.

**Reward Learning from Suboptimality.** Standard PBRL methods typically require extensive manual labeling of trajectory pairs, posing a huge burden on human annotators. Active learning methods (Eric et al., 2007) have been used to alleviate this burden by selectively presenting the most informative preference pairs for labeling. In addition, approaches such as (Palan et al., 2019) and (Ibarz et al., 2018) enhance the quality of preference data by incorporating constraints from demonstrations. Recently, methods have emerged to fully automate preference generation, reducing or eliminating the need for manual annotation. D-REX (Brown et al., 2020) was the first to inject uniformly random noise at varying levels into the BC policy to generate preferences for reward learning. LERP (Huo et al., 2023) enhances reward learning from preferences in D-REX by introducing noise directly into the reward space, rather than the action space. SSRR (Chen et al., 2021) employs AIRL for policy training, using a sigmoid function to enable reward regression. What distinguishes APEC from these existing methods is its capacity to produce preference data with much broader coverage by leveraging the diverse policies produced in the adversarial training process, enabling more robust reward learning.

## 6. Discussions and Limitations

In this paper, we introduce APEC, a framework for automated preference generation with enhanced coverage for reward learning. Building on the theoretical and empirical evidence that AIL generates policy pairs with clear preferences and good coverage, APEC selects policies from different stages of the AIL training process to produce preference data that is both diverse and accurate. This approach addresses key limitations in existing preference generation methods, which often struggle to generate sufficiently coverage samples, limiting the generalizability of learned reward models. Our experimental results on vector-based and pixel-based control tasks demonstrate that APEC significantly improves the coverage of generated preferences compared

to baseline methods. The reward models learned from these preferences align more closely with ground-truth rewards.

**Limitations and Future Works.** Although APEC represents significant progress, there is still considerable room for further improvement in future work. For example, APEC still requires interaction with the environment to generate preference data, making the extension of APEC to offline settings an interesting problem. Additionally, as a fully data-driven reward learning framework, APEC may still suffer from reward hacking in more complex tasks. A key challenge for future work is integrating human prior knowledge into the reward learning process to mitigate such risks.

## Acknowledgements

This work was supported by the NSFC (62495093), the NSFC Fundamental Research Program for Young Scholars (PhD Candidates) of the National Science Foundation of China (623B2049) and Jiangsu Science Foundation (BK20243039).

## Impact Statement

This paper contributes to advancing the field of reinforcement learning and reward modeling. While there are potential societal implications of this work, we believe none require specific attention or discussion here.

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

# A. Proof of Proposition 3.1

In this part, we present the proof of Proposition 3.1. First, Algorithm 1 outlines the procedure of a standard adversarial imitation learning algorithm.

---

**Algorithm 1** Adversarial Imitation Learning

---

**Require:** Initialized reward $r^1$, initialized policy $\widehat{\pi}^1$, reward step size $\eta^r = \sqrt{|\mathcal{S}||\mathcal{A}|/(4H^2K)}$, policy step size $\eta^\pi = \sqrt{(2\ln(|\mathcal{A}|))/(H^2K)}$.

    **for** $k = 1, 2, \ldots, K$ **do**

        Rollout $\widehat{\pi}^k$ to compute the state-action distribution $d^{\widehat{\pi}^k}$.

        Update the reward by projected gradient descent:

$$r^{k+1} = \mathcal{P}_\mathcal{R}\left(r^k - \eta^r \nabla f^k(r)\right),$$

        where $f^k(r) = V^{\widehat{\pi}^k, r} - \widehat{V}^{\pi^O, r} = H\sum_{(s,a)\in\mathcal{S}\times\mathcal{A}} r(s,a)\left(d^{\widehat{\pi}^k}(s,a) - \widehat{d^{\pi^O}}(s,a)\right)$ and $\widehat{d^{\pi^O}}$ is the empirical state-action distribution based on $\mathcal{D}^O$.

        Calculate the Q-value function $\{Q_h^{\widehat{\pi}^k, r^k}\}_{h=1}^H$ for policy $\widehat{\pi}^k$.

        Update the policy by KL-regularized policy optimization:

$$\widehat{\pi}_h^{k+1}(\cdot|s) = \max_{p\in\Delta(\mathcal{A})} \mathbb{E}_{a\sim p(\cdot)}\left[Q_h^{\widehat{\pi}^k, r^k}(s,a)\right] - \frac{1}{\eta^\pi} D_{\mathrm{KL}}\left(p(\cdot), \widehat{\pi}_h^k(\cdot|s)\right).$$

    **end for**

**Ensure:** $\pi^K$ sampled uniformly from $\{\widehat{\pi}^1, \ldots, \widehat{\pi}^K\}$.

---

Now we proceed to present the proof of Proposition 3.1. First of all, we have the following error decomposition.

$$
\begin{aligned}
V^{\pi^O, r^\star} - V^{\pi^K, r^\star} &\leq \max_{r\in\mathcal{R}} V^{\pi^O, r} - V^{\pi^K, r} \\
&\overset{(a)}{=} \frac{1}{K}\max_{r\in\mathcal{R}}\sum_{k=1}^K V^{\pi^O, r} - V^{\widehat{\pi}^k, r} \\
&= \frac{1}{K}\underbrace{\left(\max_{r\in\mathcal{R}}\sum_{k=1}^K V^{\pi^O, r} - V^{\widehat{\pi}^k, r} - \sum_{k=1}^K V^{\pi^O, r^k} - V^{\widehat{\pi}^k, r^k}\right)}_{\text{Term I}} + \underbrace{\frac{1}{K}\sum_{k=1}^K V^{\pi^O, r^k} - V^{\widehat{\pi}^k, r^k}}_{\text{Term II}}.
\end{aligned}
$$

Here $\mathcal{R} = \{r : \mathcal{S}\times\mathcal{A} \to [0,1]\}$ is the set of reward functions. Here equation $(a)$ follows the definition of $\pi^K$. For Term I, we have that

$$
\begin{aligned}
&\frac{1}{K}\left(\max_{r\in\mathcal{R}}\sum_{k=1}^K V^{\pi^O, r} - V^{\widehat{\pi}^k, r} - \sum_{k=1}^K V^{\pi^O, r^k} - V^{\widehat{\pi}^k, r^k}\right) \\
&= \frac{1}{K}\left(\left(\max_{r\in\mathcal{R}}\sum_{k=1}^K \widehat{V}^{\pi^O, r} - V^{\widehat{\pi}^k, r} + V^{\pi^O, r} - \widehat{V}^{\pi^O, r}\right) - \sum_{k=1}^K \widehat{V}^{\pi^O, r^k} - V^{\widehat{\pi}^k, r^k} + V^{\pi^O, r^k} - \widehat{V}^{\pi^O, r^k}\right) \\
&\leq \frac{1}{K}\left(\left(\max_{r\in\mathcal{R}}\sum_{k=1}^K \widehat{V}^{\pi^O, r} - V^{\widehat{\pi}^k, r}\right) - \sum_{k=1}^K \widehat{V}^{\pi^O, r^k} - V^{\widehat{\pi}^k, r^k}\right) + 2\max_{r\in\mathcal{R}} V^{\pi^O, r} - \widehat{V}^{\pi^O, r} - \frac{1}{K}\sum_{k=1}^K V^{\pi^O, r^k} - \widehat{V}^{\pi^O, r^k} \\
&\leq \frac{1}{K}\left(\left(\max_{r\in\mathcal{R}}\sum_{k=1}^K \widehat{V}^{\pi^O, r} - V^{\widehat{\pi}^k, r}\right) - \sum_{k=1}^K \widehat{V}^{\pi^O, r^k} - V^{\widehat{\pi}^k, r^k}\right) + 4\max_{r\in\mathcal{R}}\left|V^{\pi^O, r} - \widehat{V}^{\pi^O, r}\right|.
\end{aligned}
$$

Here $\widehat{V}^{\pi^O, r} = (1/|\mathcal{D}^E|)\sum_{\tau\in\mathcal{D}^E}\sum_{h=1}^H r_h(\tau(s_h), \tau(a_h))$ is the estimated policy value for policy $\pi^O$ regarding reward $r$.

For the first term in RHS, we have that

$$\max_{r \in \mathcal{R}} \sum_{k=1}^{K} \widehat{V}^{\pi^O,r} - V^{\widehat{\pi}^k,r} - \left( \sum_{k=1}^{K} \widehat{V}^{\pi^O,r^k} - V^{\widehat{\pi}^k,r^k} \right) = \sum_{k=1}^{K} V^{\widehat{\pi}^k,r^k} - \widehat{V}^{\pi^O,r^k} - \min_{r \in \mathcal{R}} \sum_{k=1}^{K} V^{\widehat{\pi}^k,r} - \widehat{V}^{\pi^O,r},$$

which is exactly the regret for solving the online optimization problem with loss functions $f^k(r) = V^{\widehat{\pi}^k,r} - \widehat{V}^{\pi^O,r}$. We leverage the following lemma to upper bound this regret.

**Lemma A.1.** *Consider Algorithm 1, we have*

$$\sum_{k=1}^{K} f^k\left(r^k\right) - \min_{r \in \mathcal{R}} \sum_{k=1}^{K} f^k(w) \leq 2H\sqrt{|\mathcal{S}||\mathcal{A}|K},$$

*where $f^k(r) = H \sum_{(s,a) \in \mathcal{S} \times \mathcal{A}} r(s,a)(d^{\widehat{\pi}^k}(s,a) - \widehat{d^{\pi^O}}(s,a))$.*

*Proof of Lemma A.1.* Lemma A.1 is a direct consequence of the regret bound of online gradient descent (Shalev-Shwartz, 2012). To apply such a regret bound, we need to verify that 1) the iterate norm $\|r\|_2$ has an upper bound; 2) the gradient norm $\|\nabla_r f^k(r)\|_2$ also has an upper bound. The first point is easy to show, i.e., $\|r\|_2 \leq \sqrt{|\mathcal{S}||\mathcal{A}|}$ by the condition that $r \in \mathcal{R} = \{r : \mathcal{S} \times \mathcal{A} \to [0,1]\}$. For the second point, we have

$$
\begin{aligned}
\left\| \nabla_r f^k(r) \right\|_2 &= H \sqrt{\sum_{(s,a) \in \mathcal{S} \times \mathcal{A}} \left( d^{\widehat{\pi}^k}(s,a) - \widehat{d^{\pi^O}}(s,a) \right)^2} \\
&\leq H \sqrt{\sum_{(s,a) \in \mathcal{S} \times \mathcal{A}} \left( d^{\widehat{\pi}^k}(s,a) \right)^2 + \left( \widehat{d^{\pi^O}}(s,a) \right)^2} \\
&\leq \sqrt{2}H.
\end{aligned}
$$

Invoking Corollary 2.7 in (Shalev-Shwartz, 2012) with $B = \sqrt{|\mathcal{S}||\mathcal{A}|}$, $L = \sqrt{2}H$ and $\eta^r = B/(L\sqrt{2K}) = \sqrt{|\mathcal{S}||\mathcal{A}|}/(2H\sqrt{K})$ finishes the proof. $\square$

Lemma A.1 demonstrates that

$$\max_{r \in \mathcal{R}} \sum_{k=1}^{K} \widehat{V}^{\pi^O,r} - V^{\widehat{\pi}^k,r} - \left( \sum_{k=1}^{K} \widehat{V}^{\pi^O,r^k} - V^{\widehat{\pi}^k,r^k} \right) \leq 2H\sqrt{|\mathcal{S}||\mathcal{A}|K}.$$

Furthermore, we have that

$$
\begin{aligned}
4\max_{r \in \mathcal{R}} \left| V^{\pi^O,r} - \widehat{V}^{\pi^O,r} \right| &= 4\max_{r \in \mathcal{R}} H \left| \sum_{(s,a) \in \mathcal{S} \times \mathcal{A}} \left( d^{\pi^O}(s,a) - \widehat{d^{\pi^O}}(s,a) \right) r(s,a) \right| \\
&\leq 4H \left\| d^{\pi^O}(\cdot, \cdot) - \widehat{d^{\pi^O}}(\cdot, \cdot) \right\|_1.
\end{aligned}
$$

According to the concentration inequality for $\ell_1$-norm (Weissman et al., 2003; Kang et al., 2023), with probability at least $1 - \delta$, we have

$$\left\| d^{\pi^O}(\cdot, \cdot) - \widehat{d^{\pi^O}}(\cdot, \cdot) \right\|_1 \leq \sqrt{\frac{2|\mathcal{S}||\mathcal{A}| \ln(1/\delta)}{|\mathcal{D}^O|}}.$$

Consequently, for Term I, we have that

$$\frac{1}{K} \left( \max_{r \in \mathcal{R}} \sum_{k=1}^{K} V^{\pi^O, r} - V^{\widehat{\pi}^k, r} - \sum_{k=1}^{K} V^{\pi^O, r^k} - V^{\widehat{\pi}^k, r^k} \right)$$

$$\leq \frac{1}{K} \left( \left( \max_{r \in \mathcal{R}} \sum_{k=1}^{K} \widehat{V}^{\pi^O, r} - V^{\widehat{\pi}^k, r} \right) - \sum_{k=1}^{K} \widehat{V}^{\pi^O, r^k} - V^{\widehat{\pi}^k, r^k} \right) + 4 \max_{r \in \mathcal{R}} \left| V^{\pi^O, r} - \widehat{V}^{\pi^O, r} \right|$$

$$\leq 2H \sqrt{\frac{|\mathcal{S}||\mathcal{A}|}{K}} + 4H \sqrt{\frac{2|\mathcal{S}||\mathcal{A}| \ln(1/\delta)}{|\mathcal{D}^O|}}.$$

For Term II, according to the policy difference lemma, we have that

$$\frac{1}{K} \sum_{k=1}^{K} V^{\pi^O, r^k} - V^{\widehat{\pi}^k, r^k} = \frac{1}{K} \sum_{k=1}^{K} \mathbb{E} \left[ \sum_{h=1}^{H} \langle \pi_h^O(\cdot|s_h) - \widehat{\pi}_h^k(\cdot|s_h), Q_h^{\widehat{\pi}^k, r^k}(s_h, \cdot) \rangle \Big| \pi^O \right].$$

For each fixed $(s, h) \in \mathcal{S} \times [H]$, we analyze the term

$$\sum_{k=1}^{K} \langle \pi_h^O(\cdot|s) - \widehat{\pi}_h^k(\cdot|s), Q_h^{\widehat{\pi}^k, r^k}(s, \cdot) \rangle = \sum_{k=1}^{K} \langle \widehat{\pi}_h^k(\cdot|s), -Q_h^{\widehat{\pi}^k, r^k}(s, \cdot) \rangle - \langle \pi_h^O(\cdot|s), -Q_h^{\widehat{\pi}^k, r^k}(s, \cdot) \rangle,$$

which is exactly the regret of online linear minimization with respect to loss function $f^k(p) = \sum_{a \in \mathcal{A}} p(a) Q_h^{\widehat{\pi}^k, r^k}(s, a)$ for probability distribution $p \in \Delta(\mathcal{A})$. From the perspective of online linear minimization, we can view the policy update as the well-known mirror descent update. Based on the theory of online mirror descent (Shalev-Shwartz, 2007), we have that

$$\sum_{k=1}^{K} \langle \widehat{\pi}_h^k(\cdot|s), -Q_h^{\widehat{\pi}^k, r^k}(s, \cdot) \rangle - \langle \pi_h^O(\cdot|s), -Q_h^{\widehat{\pi}^k, r^k}(s, \cdot) \rangle = \frac{\log(|\mathcal{A}|)}{\eta^\pi} + \frac{\eta^\pi}{2} \sum_{k=1}^{K} \sum_{a \in \mathcal{A}} \pi_h^k(a \mid s) \left( Q_h^{\widehat{\pi}^k, r^k}(s, a) \right)^2$$

$$\leq \frac{\ln(|\mathcal{A}|)}{\eta^\pi} + \frac{\eta^\pi H^2 K}{2}$$

$$= \sqrt{2 \ln(|\mathcal{A}|) H^2 K}.$$

The last equation holds by choosing $\eta^\pi = \sqrt{(2 \ln(|\mathcal{A}|))/(H^2 K)}$. Consequently, we have the following upper bound on Term II.

$$\frac{1}{K} \sum_{k=1}^{K} V^{\pi^O, r^k} - V^{\widehat{\pi}^k, r^k} = \frac{1}{K} \sum_{k=1}^{K} \mathbb{E} \left[ \sum_{h=1}^{H} \langle \pi_h^O(\cdot|s_h) - \widehat{\pi}_h^k(\cdot|s_h), Q_h^{\widehat{\pi}^k, r^k}(s_h, \cdot) \rangle \Big| \pi^O \right]$$

$$\leq 2H^2 \sqrt{\frac{2 \ln(|\mathcal{A}|)}{K}}.$$

Combining the upper bounds on Term I and Term II yields that

$$V^{\pi^O, r^\star} - V^{\pi^K, r^\star} \leq 2H \sqrt{\frac{|\mathcal{S}||\mathcal{A}|}{K}} + 4H \sqrt{\frac{2|\mathcal{S}||\mathcal{A}| \ln(1/\delta)}{|\mathcal{D}^O|}} + 2H^2 \sqrt{\frac{2 \ln(|\mathcal{A}|)}{K}}$$

$$\leq 4H^2 \sqrt{\frac{2|\mathcal{S}||\mathcal{A}| \ln(|\mathcal{A}|)}{K}} + 4H \sqrt{\frac{2|\mathcal{S}||\mathcal{A}| \ln(1/\delta)}{|\mathcal{D}^O|}}.$$

We complete the proof.

## B. Implementation Details

### B.1. Implementation Details of APEC

**Policy Collection.** We choose DAC (Kostrikov et al., 2019), an AIL algorithm known for its sample efficiency and excellent convergence performance, to collect policies. Moreover, we use SAC (Haarnoja et al., 2018) for policy updates in

vector-based tasks and DrQ-v2 (Yarats et al.) for pixel-based tasks, as outlined in Algorithm 1. Specifically, for vector-based MuJoCo tasks, we train policies using a single suboptimal demonstration for 400 iterations with 5000 steps per iteration. Checkpoints of the policy model are saved at the end of each iteration, labeled by the corresponding iteration number. For pixel-based DMControl tasks, we train a policy with 10 suboptimal demonstrations. The training spans 4M frames, with evaluations conducted every 20,000 frames. During each evaluation phase, a checkpoint of the policy is saved. We select the official codes of PAIL(Cao et al., 2024) and ROT(Haldar et al., 2023) as the codebase for Mujoco and DMControl tasks respectively and use their recommended default hyperparameters as listed in Table 5 and 6.

| Hyperparameter | Value |
| --- | --- |
| Hidden layers | 2 |
| Hidden dimension | 128 |
| Auto Alpha | False |
| Activation | ReLU |
| Batch size | 256 |
| Buffer size | 1000000 |
| Gradient penalty coefficient | 0 |
| Exploration steps | 10000 |
| Learning rate | 1e-3 |
| Optimizer | Adam |

*Table 5.* Hyperparameters of AIL in Mujoco tasks.

| Hyperparameter | Value |
| --- | --- |
| Hidden layers | 2 |
| Feature dimension | 50 |
| Hidden dimension | 1024 |
| Activation | ReLU |
| Batch size | 256 |
| Buffer size | 150000 |
| Gradient penalty coefficient | 10 |
| Exploration steps | 2000 |
| DDPG exploration schedule | linear(1,0.1,500000) |
| Learning rate | 1e-4 |
| Optimizer | Adam |

*Table 6.* Hyperparameters of AIL in DMControl tasks.

**Preference Generation.** In each iteration, we sample policy pairs $\pi_i$ and $\pi_j$ from the policy set and rollout trajectories, only trajectory pairs that satisfy both the epoch criterion and the Wasserstein distance criterion conditions are added to the preference buffer. Table 7 lists the hyperparameters used in this process. For Mujoco tasks, we collect 1,000 preference pairs, while for more challenging DMControl tasks, we collect 5,000 pairs.

| Task | epoch threshold $k$ | distance threshold $\delta$ |
| --- | --- | --- |
| Mujoco | 10 | 0.1 |
| walker_run | 10 | 0.01 |
| walker_walk | 15 | 0 |
| cheetah_run | 15 | 0 |

*Table 7.* Hyperparameters of preference generation.

**Reward Learning.** For Mujoco tasks, the reward is trained for 100 epochs, each using 1,000 samples. For DMControl tasks, all samples in the preference buffer are used in each iteration and are trained for 50 epochs. In addition, we implement a random shift augmentation for pixel-based data as recommended in DrQ-v2 to improve the robustness of the reward by encouraging the model to learn invariant features under various shifts in the input. The hyperparameters of reward learning are listed in Table 8 and 9.

**Reward Evaluation.** We evaluate the reward by using it to train RL agents with the same policy learning algorithms applied in the policy collection stage (SAC for Mujoco and DrQ-v2 for DMControl). We initialize the buffer with selected trajectories from the AIL buffer in the policy collection stage to speed up convergence. For Mujoco tasks, 10 trajectories are chosen, while for DMControl tasks, all available trajectories are utilized. Both this stage and the Policy Collection stage share the same hyperparameters for policy learning.

## C. Additional Experimental Results

### C.1. Reward Accuracy

We visualized the return correlation and the reward correlation of across all tasks, as shown in Figures 6 and Figure 7. Here, we only present the results for seed=1 due to space constraints.

| Hyperparameter | Value |
|---|---|
| Hidden layers | 2 |
| Hidden dimension | 512 |
| Activation | ReLU |
| Batch size | 256 |
| Weight decay | 1e-3 |
| Learning rate | 1e-4 |
| Optimizer | Adam |

*Table 8.* Hyperparameters of reward model in Mujoco tasks.

| Hyperparameter | Value |
|---|---|
| Hidden layers | 2 |
| Feature dimension | 50 |
| Hidden dimension | 1024 |
| Activation | ReLU |
| Batch size | 256 |
| Weight decay | 1e-4 |
| Learning rate | 1e-4 |
| Optimizer | Adam |

*Table 9.* Hyperparameters of reward model in DMControl tasks.

### C.2. Generated Preference Accuracy

The generated preference accuracy is summarized in Table 10.

| Environment | SSRR | D-REX | APEC(ours) |
|---|---|---|---|
| Ant | $0.89_{\pm 0.03}$ | $\mathbf{0.96}_{\pm 0.01}$ | $0.94_{\pm 0.01}$ |
| HalfCheetah | $0.99_{\pm 0.01}$ | $0.94_{\pm 0.02}$ | $\mathbf{1.00}_{\pm 0.00}$ |
| Hopper | $0.96_{\pm 0.03}$ | $0.89_{\pm 0.02}$ | $\mathbf{0.99}_{\pm 0.00}$ |
| Humanoid | $0.95_{\pm 0.03}$ | $0.31_{\pm 0.17}$ | $\mathbf{0.99}_{\pm 0.00}$ |
| Walker2d | $0.88_{\pm 0.05}$ | $0.70_{\pm 0.21}$ | $\mathbf{0.99}_{\pm 0.00}$ |
| Average | 0.93 | 0.76 | **0.98** |
| cheetah_run | $0.91_{\pm 0.00}$ | $0.94_{\pm 0.00}$ | $\mathbf{0.98}_{\pm 0.00}$ |
| walker_run | $0.93_{\pm 0.00}$ | $0.72_{\pm 0.02}$ | $\mathbf{1.00}_{\pm 0.00}$ |
| walker_walk | $0.89_{\pm 0.00}$ | $0.89_{\pm 0.01}$ | $\mathbf{0.95}_{\pm 0.01}$ |
| Average | 0.91 | 0.85 | **0.98** |

*Table 10.* Generate accuracy on Mujoco and DMControl tasks over 5 random seeds.

### C.3. Additional experiment on Atari

Table 11 shows an additional experiment on the Atari game Pong, which features a discrete action space. Results show that APEC consistently outperforms both DREX and LERP in terms of task performance, demonstrating its broader applicability. Notably, for the Pong experiment, we used an alternative AIL method, IQ-Learn (Garg et al., 2021), to generate preference data, which further highlights APEC's generalizability across different AIL algorithms.

*Table 11.* Task performance comparison on Pong

| Method | Demo Mean | Task Performance |
|---|---|---|
| DREX | 3.7 | -9.5 |
| LERP | 3.7 | -20.0 |
| APEC (ours) | 3.7 | -0.44 |

### C.4. Sensitivity analysis on hyperparameters

As shown in Table 12, 13 and 14, We perform ablation studies on key hyperparameter choices, including the **training epoch interval**, **Wasserstein distance threshold**, and **segment length for reward learning**, across both feature-based `HalfCheetah` and pixel-based `walker_run` tasks. The "default" refers to the values used in our original experiments. The results show that APEC is not sensitive to different hyperparameter choices.

*Table 12.* Ablation study on training epoch interval

| Task | 5 | 10 (default) | 50/20 |
|---|---|---|---|
| HalfCheetah | 11,957 | **12,232** | 10,019 |
| walker_run | **710** | 701 | 591 |

*Table 13.* Ablation study on Wasserstein distance threshold (higher is better)

| Task | Lower | Default | Higher |
|---|---|---|---|
| HalfCheetah (0.05/0.1/0.2) | **13,291** | 12,232 | 8,848 |
| walker_run (0.005/0.01/0.02) | 709 | 701 | **736** |

*Table 14.* Ablation study on segment length (higher is better)

| Task | Shorter | Default | Longer |
|---|---|---|---|
| HalfCheetah (100/500/1000) | 12,658 | **13,434** | 12,232 |
| walker_run (5/10/20) | **712** | 701 | 696 |

*Table 15.* Experiments on four MuJoCo tasks under two conditions: lower quality demonstrations and more demonstrations. In each value pair "x/y", x represents the performance of the demonstrations, while y is the performance of the policy trained with the reward model learned from APEC.

| Performance | Ant | HalfCheetah | Hopper | Walker2d |
|---|---|---|---|---|
| low 1 demo | 2413 / 1522 | 4232 / 7874 | 1325 / 3300 | 2205 / 2089 |
| high 1 demo | 4105 / 5221 | 7313 / 12232 | 2275 / 3310 | 3706 / 5394 |
| high 10 demo | 4179 / 4830 | 7385 / 14700 | 2388 / 3325 | 3728 / 5390 |

## C.5. Reward Hacking Analysis

Figure 8, 9, 10, 11, and 12 demonstrates the relationship between the predicted return and ground truth return for the baselines and APEC. The learned return of all algorithms increases with the number of training steps. However, only APEC ensures that the ground truth return increases throughout the process, while the other methods show a decline in ground truth return, indicating reward hacking. These plots show the strong robustness and generalizability of APEC's learned reward.

## D. Computational Resources

All experiments were performed on an RTX 4090 GPU platform. In Stage 1 (Policy Collection), Mujoco tasks require approximately 24 hours to complete 2 million training steps, while DMControl tasks take about 10 hours to complete 1 million frames. Stage 2 (Preference Generation) does not require GPU resources. In Stage 3 (Reward Model Training), Mujoco tasks typically take around 20 minutes, while DMControl tasks require approximately 3 hours.

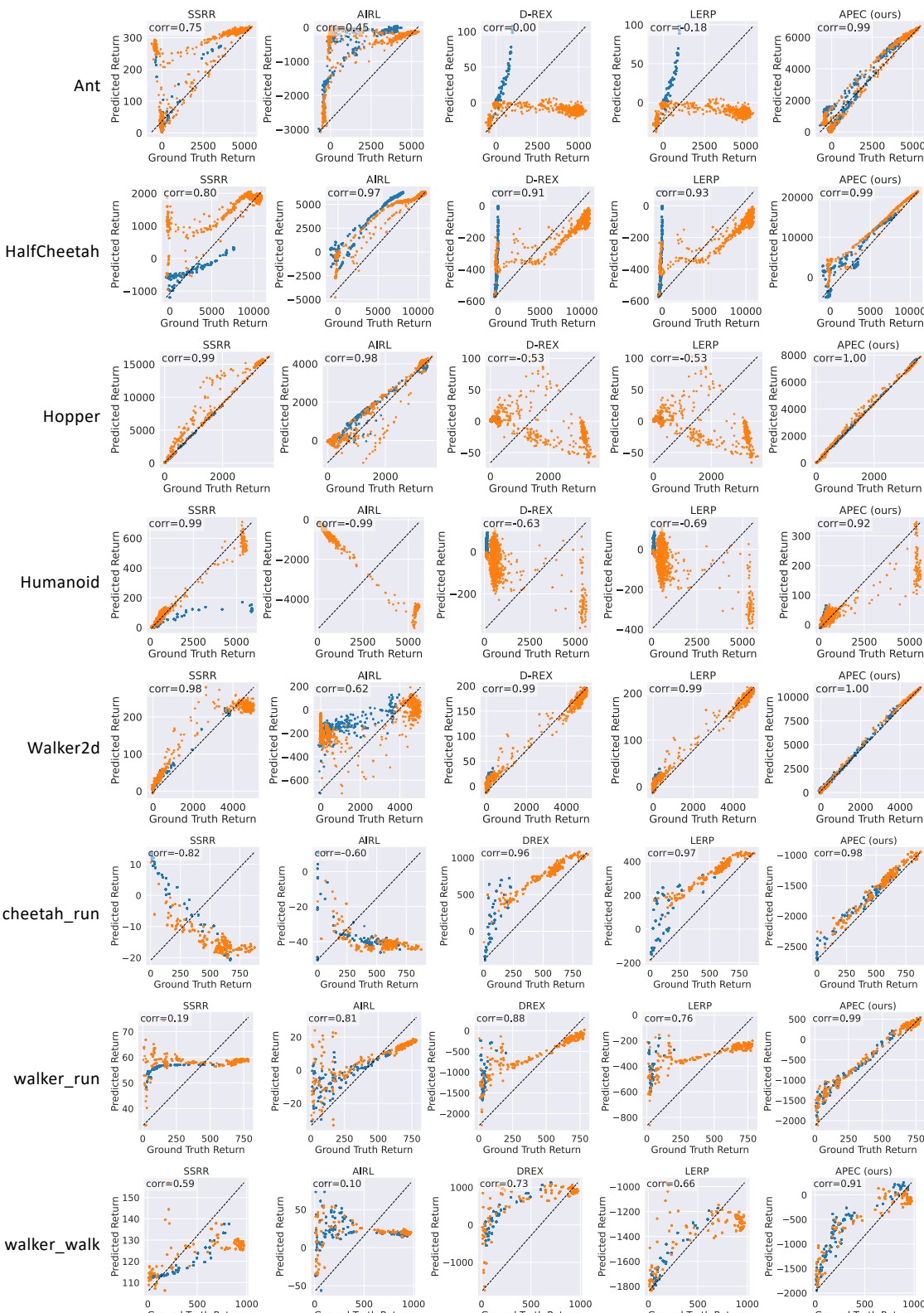

*Figure 6.* Return correlation visualization. Blue dots represent trajectories generated for training reward models, and orange dots represent additional trajectories not seen during training. Here, the x-axis represents the ground truth return, while the y-axis represents the return predicted by the learned reward models.

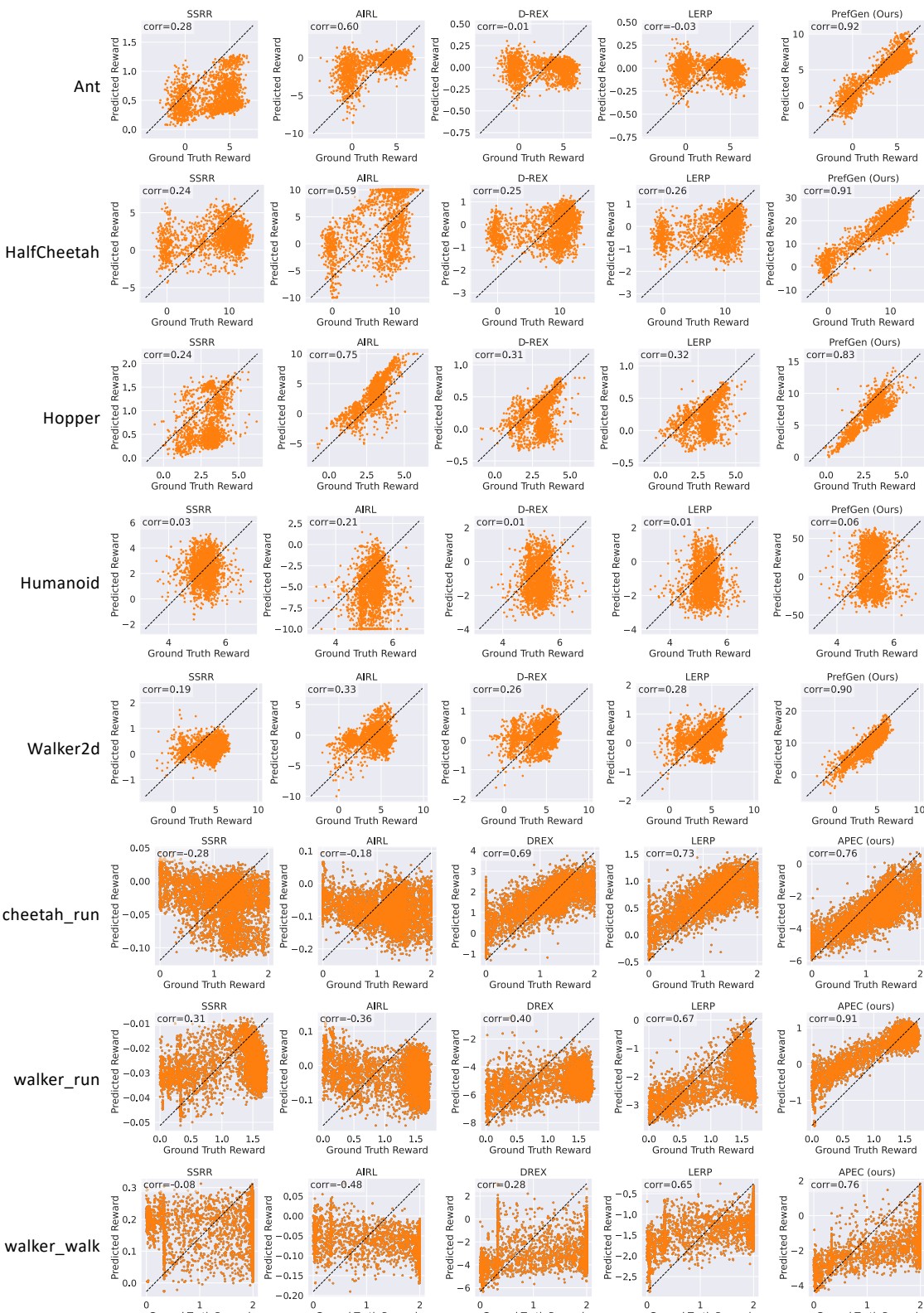

*Figure 7.* Reward correlation Return correlation visualization. Orange dots represent state-action pairs on the test dataset. Here, the x-axis represents the ground truth reward, while the y-axis represents the reward predicted by the learned reward models.

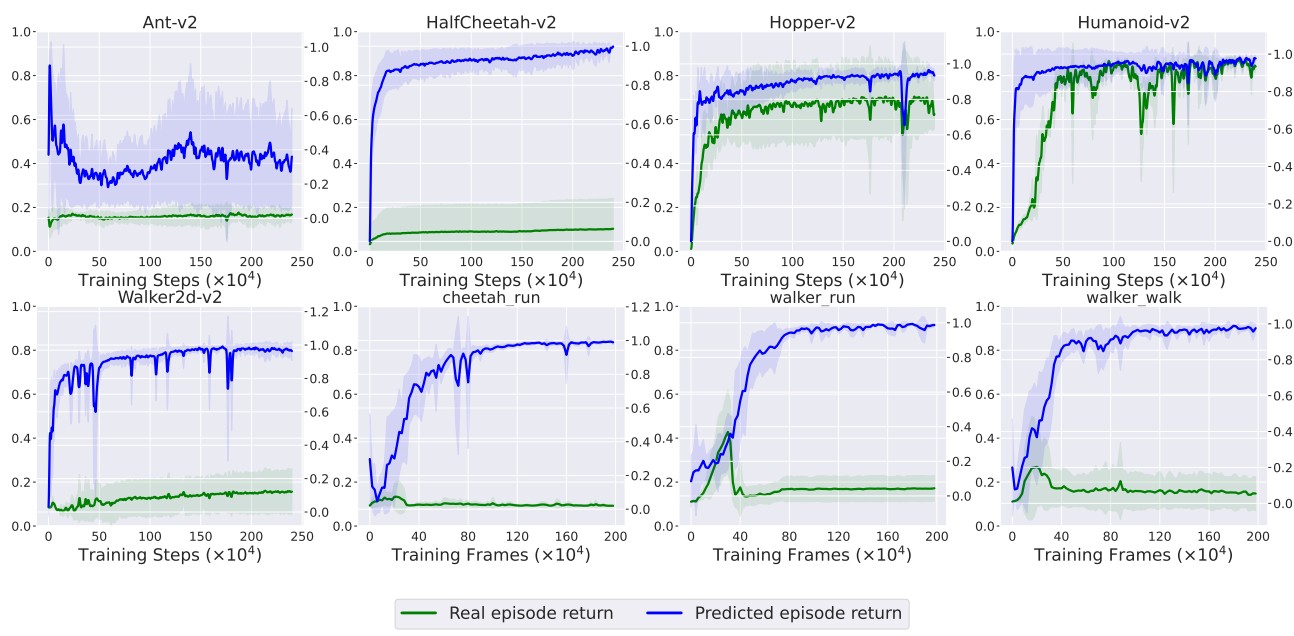

*Figure 8.* The relationship between the predicted return and the ground truth return in SSRR. The left y-axis represents the normalized episode return, while the right y-axis also indicates the normalized episode return for comparison. The x-axis denotes the number of training steps.

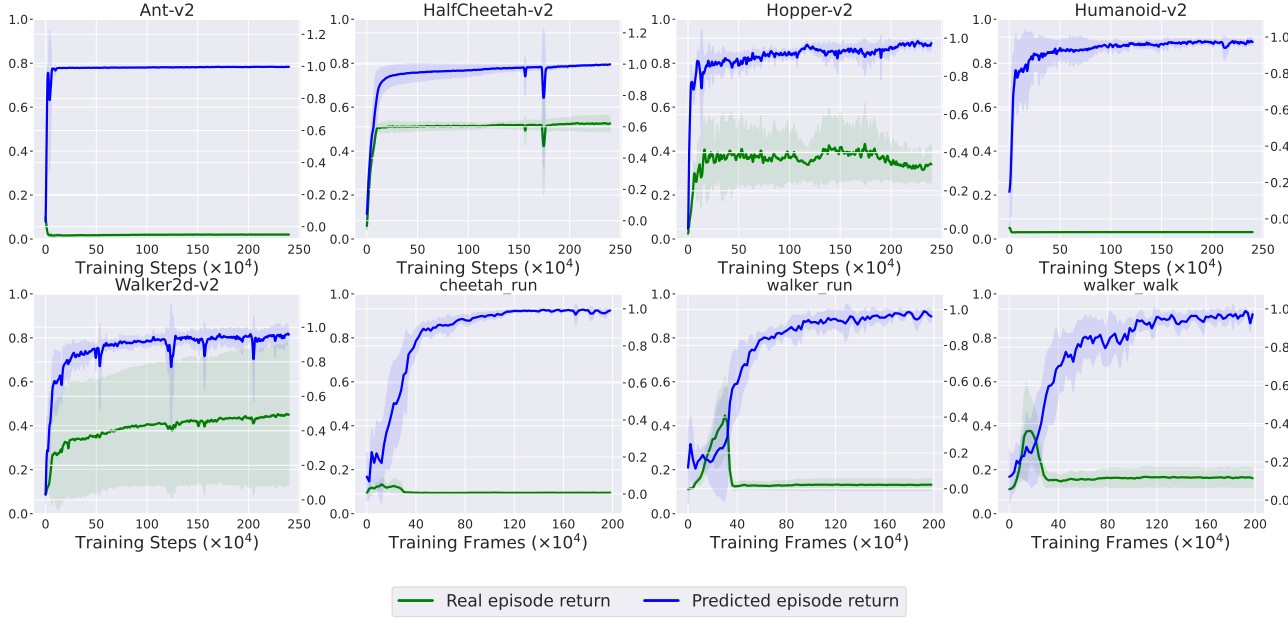

*Figure 9.* The relationship between the predicted return and the ground truth return in AIRL. The left y-axis represents the normalized episode return, while the right y-axis also indicates the normalized episode return for comparison. The x-axis denotes the number of training steps.

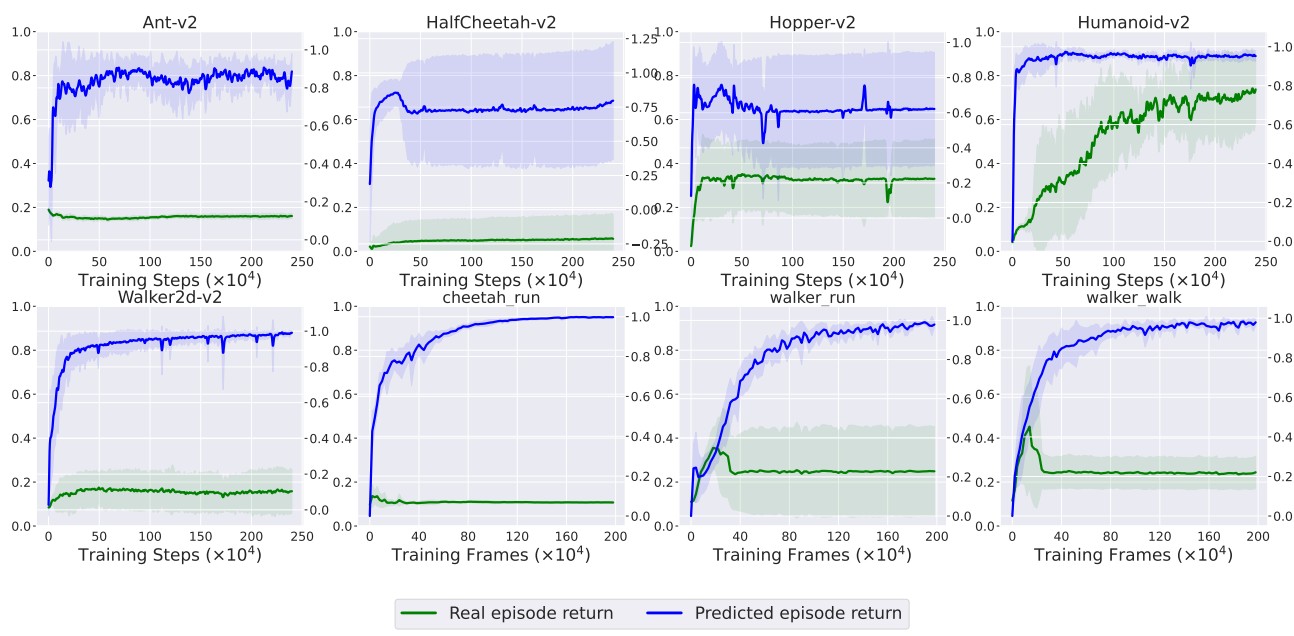

*Figure 10.* The relationship between the predicted return and the ground truth return in D-REX. The left y-axis represents the normalized episode return, while the right y-axis also indicates the normalized episode return for comparison. The x-axis denotes the number of training steps.

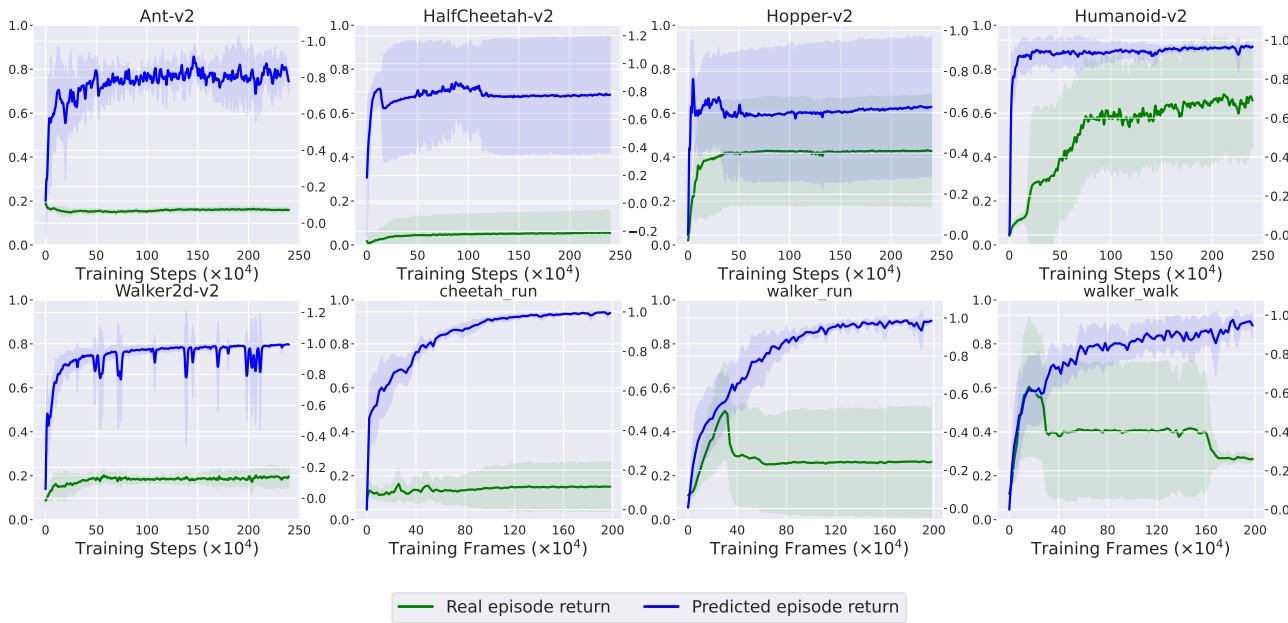

*Figure 11.* The relationship between the predicted return and the ground truth return in LERP. The left y-axis represents the normalized episode return, while the right y-axis also indicates the normalized episode return for comparison. The x-axis denotes the number of training steps.

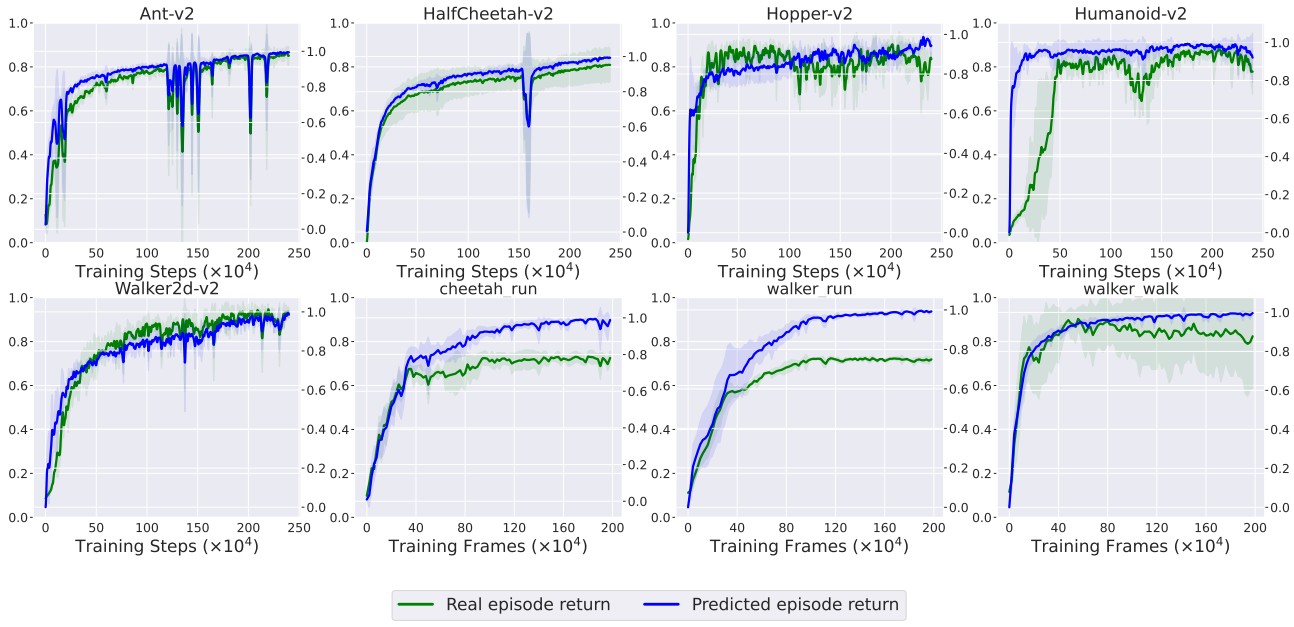

*Figure 12.* The relationship between the predicted return and the ground truth return in APEC. The left y-axis represents the normalized episode return, while the right y-axis also indicates the normalized episode return for comparison. The x-axis denotes the number of training steps.

