# OpenReview forum: "Improving Reward Model Generalization from Adversarial Process Enhanced Preferences"
_ICML.cc/2025/Conference — ICML 2025 poster_

### Official Review · Reviewer_LmzP · 2025-03-02

**Overall Recommendation:** 4

**Summary:**

The paper presents APEC, a novel approach to improving reward model learning by generating diverse preference pairs from different stages of adversarial imitation learning. This ensures broader coverage of the learning process. Theoretical analysis shows that later AIL policies are generally preferred over earlier ones. Experiments on both vector-based and pixel-based control tasks demonstrate that APEC outperforms existing methods, leading to reward models that better align with ground-truth rewards and produce stronger RL policies.

**Claims And Evidence:**

Yes, the claims in this paper are supported by theoretical analysis and empirical experiments.

**Essential References Not Discussed:**

The adversarial imitation learning framework is closely related to GAN training, where many techniques have been proposed to improve stability. I suggest discussing these stabilization techniques in the appendix [1][2]. In particular, [2] explores leveraging previous discriminators and generators to enhance GAN training, which share similar idea with this paper.


[1] Stabilizing generative adversarial networks: A survey[J]. arXiv preprint arXiv:1910.00927, 2019.

[2] Training gans with historical models[C]//Proceedings of the European Conference on Computer Vision (ECCV). 2018: 119-134.

**Experimental Designs Or Analyses:**

Yes.

**Methods And Evaluation Criteria:**

Yes, Mujoco and DMControl tasks are common benchmarks in this domain.

**Other Comments Or Suggestions:**

None.

**Other Strengths And Weaknesses:**

The paper presents a meaningful contribution by improving reward generalization through enhanced preference coverage, supported by empirical and theoretical validation.

**Strengths:**
- APEC integrates adversarial imitation learning with automated preference generation. The theoretical analysis of policy value progression in AIL further grounds the method’s design.
- The studied problem is a critical challenge in reward learning.
-  The presentation is clear and well-organized.

**Weaknesses:**
- The method is dependent on that later policies are better than earlier ones, but this may not always hold due to training fluctuations. A discussion on how this randomness impacts learning would be valuable.
- In Eq. 1, the loss calculation uses a section of the trajectory, but the impact of the parameter l on performance is not explored.
- Comparisons are limited to older methods (e.g., D-REX, AIRL, SSRR), and recent advances in preference learning or AIL variants after 2023 are not included. For instance, [1] is highly relevant for learning generalizable reward models. Additionally, including a baseline with an oracle reward would help illustrate the gap to the upper bound.
- Limited discussion with related domains: The adversarial imitation learning framework shares similarities with GAN training, yet there is minimal discussion of techniques from this domain. Furthermore, the work is closely tied to preference learning in LLMs, and a discussion of related works, such as [2][3][4], which also focus on improving reward model generalization, is recommended.


[1] BC-IRL: Learning generalizable reward functions from demonstrations. arXiv preprint arXiv:2303.16194, 2023.

[2] Reward Generalization in RLHF: A Topological Perspective. arXiv preprint arXiv:2402.10184, 2024.

[3] Regularizing Hidden States Enables Learning Generalizable Reward Model for LLMs. arXiv e-prints, 2024: arXiv: 2406.10216.

[4] Improving Discriminative Capability of Reward Models in RLHF Using Contrastive Learning. Proceedings of the 2024 Conference on Empirical Methods in Natural Language Processing. 2024: 15270-15283.

**Questions For Authors:**

Please see the weakness part.

**Relation To Broader Scientific Literature:**

I think this paper has a broader impact on various research domains, including human preference learning and understanding (related to RLHF for LLMs), and areas like GAN training. However, the paper does not discuss these broader implications in detail. I suggest expanding on this aspect to better highlight its significance across related fields.

**Theoretical Claims:**

I have not thoroughly verified Proposition 3.1, but it appears to be correct upon initial inspection.

---

> ### Author Rebuttal · Authors · 2025-04-01
>
> **Q1**: The method is dependent on that later policies are better than earlier ones, but this may not always hold due to training fluctuations. A discussion on how this randomness impacts learning would be valuable.
>
> **A1**: This randomness indeed has a negative impact on the accuracy of the generated preference. Nevertheless, APEC additionally leverages the Wasserstein distance metric to help choose the policy pair, which enhances the robustness of preference generation to randomness. Experimental results in Section 4.4 show that APEC can generate more accurate preference data compared to baselines.
>
> **Q2**: In Eq. 1, the loss calculation uses a section of the trajectory, but the impact of the parameter l on performance is not explored.
>
> **A2**: We apologize for missing this ablation study. Here we perform the ablation study on **segment length for reward learning**, across both feature-based HalfCheetah and pixel-based walker_run tasks. The "default" refers to the values used in our original experiments, and we show the overall task performance here. The findings show that APEC is **not** sensitive to this parameter.
>
> | |100|500|1000(default)|
> |---|---|---|---|
> HalfCheetah	|12658	|13434|	12232|
> | |5|10(default)|20|
> |walker_run|	712|	701|	696|
>
>
> **Q3**: Comparisons are limited to older methods. Additionally, including a baseline with an oracle reward would help illustrate the gap to the upper bound. Additionally, including a baseline with an oracle reward would help illustrate the gap to the upper bound.
>
> **A3**: To address your concern, we add two recent baselines, BC-IRL [1] and DRAIL [2] and the method with oracle reward. We use the recommended hyperparameters from the source code and slightly tune them. The task performance results are shown in the table below, where we find that APEC still achieves the best performance compared to BC-IRL and DRAIL.
>
> |Performance|	Ant	|HalfCheetah|	Hopper|	Humanoid|	Walker|
> |---|---|---|---|---|---|
> |BC-IRL|	-156721	|-48835	|1084|	97|	-4.82|
> |DRAIL	|1425	|3699	|3316	|138	|3586|
> |Oracle|	5850|	12138|	3542|	5489|	5007|
> |APEC (ours)|	5221|	12232|	3310|	4953|	5394|
>
>
> [1] BC-IRL: Learning Generalizable Reward Functions from Demonstrations. ICLR’23.
>
> [2] Diffusion-Reward Adversarial Imitation Learning. NeurIPS’24.
>
> **Q4**: Limited discussion with related domains: The adversarial imitation learning framework shares similarities with GAN training, yet there is minimal discussion of techniques from this domain. Furthermore, the work is closely tied to preference learning in LLMs, and a discussion of related works, which also focus on improving reward model generalization, is recommended.
>
> **A4**: Thank you for your insightful feedback. We agree that there are notable similarities between the AIL framework and GAN training because both of them involve an adversarial learning procedure. We will replenish the discussion on GAN in the revision.
>
> Besides, while preference learning in LLM is indeed relevant to our work, it is crucial to highlight that APEC focuses on the automatic generation of diverse and accurate preference data through AIL, whereas the paper referenced by the reviewer centers on improving reward generalization by improving preference learning algorithms with fixed preference data. We will elaborate on these distinctions in the revised version.

---

> > ### Comment · Reviewer_LmzP · 2025-04-03
> >
> > Thanks for your response. My concerns have been addressed, and I will raise my score. Authors are expected to include the additional ablation studies, new baselines, and a broader discussion of related domains to further strengthen the paper.

---

> > > ### Author Response · Authors · 2025-04-05
> > >
> > > We are grateful that our response has addressed your concerns, and we appreciate your effort in evaluating our paper and raising the score. Your insightful suggestions are highly valued. We will incorporate the additional ablation studies, new baselines, and a more comprehensive discussion of related areas to enhance the quality of the paper.

---

### Official Review · Reviewer_aJ5F · 2025-03-11

**Overall Recommendation:** 3

**Summary:**

The approach of automatically generating preference data can reduce human intervention, but it has limited generalizability and coverage. To solve this problem, the authors propose APEC, which addresses these challenges by selecting policy pairs with significantly different iteration indices from the whole adversarial imitation learning process. As a result, the reward models trained with APEC align more closely with the ground-truth rewards and consistently generate preference data with broader coverage, demonstrating superior performance compared to existing methods.

## update after rebuttal

I agree that it can provide insights for researchers in this particular area, and maintain original recommendation, weak accept.

**Claims And Evidence:**

The authors argue that the proposed method, APEC, use pairs with different iteration indexes from the Adversarial IL training and provides wider coverages of preference data. The authors provide a clear explanation of the use of AIL, the criteria used in the process of sampling preference pairs, and these are well supported by both qualitative and quantitative results.

**Essential References Not Discussed:**

.

**Experimental Designs Or Analyses:**

The experimental design and analysis seem valid.

**Methods And Evaluation Criteria:**

Overall methods and evaluations are plausible. For example, to avoid the errors coming from occasional performance spike, this work provides a Wasserstein distance criterion. As shown in Figure 5, these additional criteria have shown to contribute to further performance improvement. Beyond performance gains, the authors also suggest through qualitative results that the preference coverage is broader compared to previous studies, which demonstrates the effectiveness of the proposed methodology. Furthermore, the selection of benchmarks and baselines (SSRR, LERO) appears to be appropriate. (but need to check whether there are more recent and suitable baselines available.)

**Other Comments Or Suggestions:**

.

**Other Strengths And Weaknesses:**

The proposed study is simple yet effective. Despite its simplicity, it appears to be supported by appropriate theoretical proofs.

**Questions For Authors:**

Regarding the graph in Figure 2, are there specific environments where the AIL process does not match the performance of the demonstrations? If so, is there an analysis of the performance loss resulting from that gap?

**Relation To Broader Scientific Literature:**

Compared to the preference-based RL using suboptimal data, there is a significant amount of research on preference-based RL under the assumption that human preference data is available. Overall, reducing human effort is the practical direction, but it would be helpful to broaden the scope a bit. For example, it would be valuable to explain whether there are other autonomous pipelines that operate without relying on human preference data, and if such approaches exist, what advantages the method using suboptimal data offers in comparison. Including this discussion would provide important insights for understanding the significance of this study.

**Theoretical Claims:**

The authors provide a theoretical analysis that the selected policy pairs provably hold preference relationships. This paper also provides a theoretical justification for using the AIL process.

---

> ### Author Rebuttal · Authors · 2025-04-01
>
> **Q1**: Need to check whether there are more recent and suitable baselines available
>
> **A1**: Thank you for your valuable suggestions. We add two more recent reward learning baselines, BC-IRL [1] and DRAIL [2], and evaluate them on the MuJoCo benchmark. We use the recommended hyperparameters from the source code and slightly tune them. The results are shown in the table below, where we find that APEC still achieves the best performance.
>
> | **Performance** | Ant | HalfCheetah | Hopper | Humanoid | Walker |
> | --- | --- | --- | --- | --- | --- |
> | BC-IRL | -156721 | -48835 | 1084 | 97 | -4.82 |
> | DRAIL | 1425 | 3699 | 3316 | 138 | 3586 |
> | APEC (ours) | 5221 | 12232 | 3310 | 4953 | 5394 |
>
> [1] BC-IRL: Learning Generalizable Reward Functions from Demonstrations. ICLR’23.
>
> [2] Diffusion-Reward Adversarial Imitation Learning. NeurIPS’24.
>
> **Q2**: It would be valuable to explain whether there are other autonomous pipelines that operate without relying on human preference data, and if such approaches exist, what advantages the method using suboptimal data offers in comparison. Including this discussion would provide important insights for understanding the significance of this study.
>
> **A2**: There are other autonomous methods that can operate without human preference data. Notably, LLM-based labeling [1][2] has recently emerged as a popular approach for generating preference data. However, the accuracy of LLM-generated labels is not always guaranteed due to the potential hallucinations of LLMs. Moreover, for vector-based tasks, LLMs are currently incapable of providing accurate labels because LLMs cannot understand such abstract vectors. In contrast, constructing preference pairs from suboptimal data offers a rather principled approach to generating accurate preference data. For instance, APEC builds on the theoretical property of AIL that the value gap bounds decrease as the iteration number increases.
>
> [1] Preference VLM: Leveraging VLMs for Scalable Preference-Based Reinforcement Learning. arXiv preprint ,2025
>
> [2] Online Preference-based Reinforcement Learning with Self-augmented Feedback from Large Language Model. arXiv preprint ,2025
>
> **Q3:** Regarding the graph in Figure 2, are there specific environments where the AIL process does not match the performance of the demonstrations? If so, is there an analysis of the performance loss resulting from that gap?
>
> **A3**: In our experiments, there is no environment where AIL fails to match the performance of the demonstrations.

---

### Official Review · Reviewer_YpdB · 2025-03-13

**Overall Recommendation:** 3

**Summary:**

In this paper, an automated preference generation method, namely APEC (Automated Preference generation with Enhanced Coverage), is proposed, which claims that broad coverage in the generated preference data can be ensured by leveraging the policy pairs with different iteration indices from the AIL training process. Experimental results on eight continuous control benchmark tasks demonstrate the effectiveness of the proposed approach.

**Claims And Evidence:**

Experiments are conducted to support that different iteration indexes from the AIL training process can generate broad preference data coverage. However, as Proposition 3.1 only shows that with the increase of the iterations, the value gap bound of the learned policy diminishes, I am not certain whether the claim that a progressive improvement in policy value over successive iterations can be achieved is correct or not. It is noticed that a Wasserstein distance criterion is utilized for doubly robust preference generation.

**Essential References Not Discussed:**

Necessary references are cited and discussed.

**Experimental Designs Or Analyses:**

Yes, the soundness/validity of the experimental designs is checked, and only several simple benchmark problems are adopted to verify the effectiveness of the proposed approach. It will be better that the proposed approach can be verified on some real-world applications.

**Methods And Evaluation Criteria:**

Yes, experimental results on several benchmark problems show the effectiveness of the proposed method.

**Other Comments Or Suggestions:**

No

**Other Strengths And Weaknesses:**

Strengths:
The paper aims to address an important issue in the literature, that to increase the preference buffer coverage to make the learned reward model to align more closely to ground-truth rewards, the presented method is novel and interesting, experimental results on several benchmark problems also show the effectiveness of the proposed method.

Weaknesses*:
1. As mentioned, it only shows with the increase of the iterations, the value gap bound of the learned policy diminishes. I am not certain whether such a claim can guarantee theoretically a board coverage of the preference data with significantly different iteration indices.
2. The baselines used are works published in 2017, 2020, 2021, and 2023 respectively, as this is a rapidly developing research area, some more recent baselines should be compared and discussed.
3. The work is only verified on some simple benchmark problems, it is noticed in the literature, most work uses these benchmarks as the testbed, but it is still welcome to show the potential benefits of the proposed approach in some real-world applications.

**Questions For Authors:**

1. Can only the Wasserstein distance criterion be utilized for the preference generation?
2. In the experiments, it is shown APEC can even achieve better-than-demo performance on 7 out of the 8 tasks, can the authors provide further explanations of the reason behind this phenomenon?
3. Different number of demonstrations are used in the experiments, namely, one suboptimal demonstration for Mujoco task and ten demonstrations for DMControl tasks. Does the number of the demonstrations has a high impact on the final performance?
4. It is noticed the Wasserstein distance criterion is utilized for the preference generation, and ablation study show the importance of this criterion in generating the preference data. But no ablation study is conduct on the role of the proposed policy collection, how about the performance of a randomly policy collection but with the combination of the Wasserstein distance criterion?
5. For other questions, please refer to the weaknesses part of the review.

**Relation To Broader Scientific Literature:**

The key contribution of the paper is to propose an automated preference generation method, which is critical in the learning of the reward models in RL.

**Theoretical Claims:**

Yes, the correctness of the Proposition 3.1 is checked. But as mentioned, it only shows with the increase of the iterations, the value gap bound of the learned policy diminishes. I am not certain whether such a claim can guarantee theoretically a board coverage of the preference data with significantly different iteration indices.

---

> ### Author Rebuttal · Authors · 2025-04-01
>
> **Q1:** Whether the claim that a progressive improvement in policy value over successive iterations can be achieved is correct or not.
>
> **A1:** We apologize for this confusing claim. Here we want to claim that Proposition 3.1 can indicate that policies from the AIL process will establish a preference relationship when their corresponding iteration indexes differ significantly. To see this, on one hand, Proposition 3.1 essentially proves a lower bound on $V^{\pi^k}$ ($V^{\pi^k} \geq f(k)$). On the other hand, due to the proximal update nature in AIL, it is easy to prove an upper bound on the value difference between $\pi^k$ and the initial policy $\pi^1$. In particular, we can prove that
> $$
> V^{\pi^k} - V^{\pi^1} \leq H^2 (\ln(|\mathcal{A}|))^{\frac{1}{4}} K^{\frac{3}{4}}.
> $$
> This provides an upper bound on $V^{\pi^k}$ ($V^{\pi^k} \leq g(k)$). By combining the lower and upper bounds, we can establish that for two policies $\pi^{k_1}$ and $\pi^{k_2}$ whose indexes $k_1$ and $k_2$ differ sufficiently such that $f(k_1) \geq g(k_2)$, the preference relationship $V^{\pi^{k_1}} \geq V^{\pi^{k_2}}$ holds. This insight inspires us to leverage the AIL training process to generate preference data.
>
> We appreciate your valuable question and will fix this claim in the revision.
>
> **Q2:** Whether such a claim can theoretically guarantee a board coverage of the preference data with significantly different iteration indices.
>
> **A2:** We would like to clarify that we do **not** claim that Proposition 3.1 can theoretically guarantee a board coverage of the preference data. Rather, it establishes that policies from the AIL training process will hold a preference relationship when their corresponding iteration indexes differ sufficiently. This provides the rationale for leveraging the AIL in our preference data construction methodology.
>
> For the coverage property, we empirically validate that the preference data constructed based on the AIL process can provide a wider coverage than previous methods; see Table 4 and Figure 4 for details. This improved coverage likely stems from AIL's inherent advantages over BC—specifically, AIL-trained policies typically achieve better values by avoiding the compounding errors that plague BC approaches. Therefore, the preference data collected by policies from the AIL could have a better coverage on high-return regions.
>
> **Q3:** Some more recent baselines should be compared and discussed.
>
> **A3**: We add a more recent baseline DRAIL [1]. We use the recommended hyperparameters from the source code. APEC still achieves a better performance than DRAIL.
>
> |**Performance**|Ant|HalfCheetah|Hopper|Humanoid|Walker|
> |---|---|---|---|---|---|
> |DRAIL|1425|3699|3316|138|3586|
> |APEC|5221|12232|3310|4953|5394|
>
> [1] Diffusion-Reward Adversarial Imitation Learning. NeurIPS’24.
>
> **Q4**: The work is only verified on some simple benchmark.
>
> **A4:** As noted by the reviewer, most works use these benchmarks as testbeds. We emphasize that our setup is more challenging compared to prior works. Specifically, we test on pixel-based control tasks and use fewer demonstrations. APEC consistently achieves strong performance across these challenging tasks, showing the potential benefits for real-world tasks.
>
>
> **Q5**: Can only the Wasserstein distance criterion be utilized for the preference generation?
>
> **A5**: We use the L2 distance as an alternative. We find that APEC with L2 distance can still have a satisfying performance.
>
> |reward corr|Ant|HalfCheetah|Hopper|Humanoid|Walker|
> |---|---|---|---|---|---|
> |L2|0.86|0.88|0.71|0.01|0.86|
> |Wasserstein|0.94|0.91|0.83|0.04|0.88|
>
> **Q6**: How about the performance of a random policy collection with the combination of the Wasserstein distance criterion?
>
> **A6**: We conduct experiments using only the Wasserstein distance criterion for constructing preference pairs. The results show that this leads to significantly lower performance across all five tasks. This further reinforces the importance of epoch interval filtering.
>
> |**reward_corr**|Ant|HalfCheetah|Hopper|Humanoid|Walker2d|
> |---|---|---|---|---|---|
> |Wasserstein|0.70|0.72|0.39|0.00|0.57|
> |APEC|0.94|0.91|0.83|0.04|0.88|
>
> **Q7**: Does the number of the demonstrations have a high impact on the final performance?
>
> **A7**: We conduct experiments with different numbers of demonstrations. The results show that the number of demonstrations has a limited impact on the final performance.
>
> |#demo|Ant|HalfCheetah|Hopper|Walker2d|
> |---|---|---|---|---|
> |1|5221|12232|3310|5394|
> |10|4830|14700|3325|5390|
>
> **Q8**: Curious on achieving better-than-demo performance.
>
> **A8**: In certain tasks with structured true reward (e.g., linear w.r.t the state feature), PBRL could identify the underlying structure, resulting in a generalizable reward. Such a generalizable reward can guide policy learning in unseen regions, thereby achieving better-than-demo performance.

---

> > ### Comment · Reviewer_YpdB · 2025-04-04
> >
> > Thanks for the authors' response and most of my concerns have been addressed. But I am still confused about the response of Q1. Under which specific conditions can we ensure f(k_1) \geq g(k_2)? In my view, only stating that indexes differ sufficiently is not enough.

---

> > > ### Author Response · Authors · 2025-04-05
> > >
> > > Thanks for your remote response! We are pleased that most of your concerns have been addressed.
> > >
> > > For your mentioned question, we can derive the specific condition by solving the inequality $f (k_1) \geq g (k_2)$. Specifically, we aim to solve the inequality of
> > >
> > > $$
> > > V^{\pi^O} - C_1 k_1^{-1/2} - \varepsilon_{\text{stat}} \geq V^{\pi^1} + C_2 k_2^{3/4}.
> > > $$
> > >
> > > Here $\pi^O$ is the policy generating the demonstrations, $\pi^1$ is the initial policy, $C_1 = 4H^2 \sqrt{2|\mathcal{S}| |\mathcal{A}| \log (|\mathcal{A}| )}, C_2 = H (\log (|\mathcal{A}| ))^{1/4} (L_r + 2HL_p)$ are coefficients depending on the properties of the MDP, $L_r$ and $L_p$ are Lipschitz constants of the reward and transition.
> > >
> > > Rearranging to isolate $k_1$, we obtain:
> > >
> > > $$
> > > k_1 \geq \frac{C_1^2}{(V^{\pi^{O}} - V^{\pi^1} - \varepsilon_{\text{stat}} - C_2 k_2^{3/4})^2 }.
> > > $$
> > >
> > > This formula allows us to determine the appropriate $k_1$ value for any given $k_2$, providing a clear condition for selecting policy pairs with clear preferences.
> > >
> > > We will replenish the above discussion in the revision. We hope that this response can address your concern satisfactorily. We would be grateful if you could re-evaluate our paper based on our responses.

---

### Official Review · Reviewer_Gguy · 2025-03-16

**Overall Recommendation:** 2

**Summary:**

APEC addresses the challenge of generating high-quality preference data for reward modeling in sequential decision-making without human expertise. Unlike previous approaches that rely on injecting random noise into fixed policies, APEC selects policy pairs with significantly different iteration indices from the adversarial imitation learning (AIL) process. The authors provide theoretical validation that later policies in the AIL process are preferred over earlier ones, ensuring that the generated preference data has broader coverage. Empirical evaluations demonstrate that APEC produces reward models that align more closely with ground-truth rewards, resulting in improved policy performance.

**Claims And Evidence:**

The paper's claim that later AIL policies are preferred over earlier ones is adequately supported through both empirical and theoretical evidence. Figure 2 clearly demonstrates increasing policy returns over training iterations across multiple environments. Proposition 3.1 provides mathematical validation by proving that the value gap between expert and learned policies decreases as iterations increase, with appropriate consideration for statistical error.

**Essential References Not Discussed:**

No significant omissions of relevant literature were identified that would impact understanding of the paper's contributions.

**Experimental Designs Or Analyses:**

The experimental design is generally sound with key strengths: diverse benchmarks (MuJoCo and DMControl), clearly defined suboptimal demonstrations (50-80% of optimal), test dataset for generalization assessment, controlled sample sizes across methods, and appropriate evaluation metrics.

Limitations include: restriction to continuous control tasks, relatively high-quality demonstrations despite claims of challenging setup, focus on correlation metrics rather than task performance, limited discussion of computational efficiency or training stability, and no exploration of APEC's performance with varying amounts of demonstration data.

**Methods And Evaluation Criteria:**

The proposed methods are well-structured and appropriate for automated preference generation. The three-component approach (collecting AIL policies, generating diverse preferences using both epoch differences and Wasserstein distance criteria, and adapting reward learning for trajectory segments) addresses both theoretical foundations and practical challenges. The Wasserstein distance criterion intelligently handles trajectory variance and performance spikes, while the segment-based approach to reward learning sensibly adapts the Bradley-Terry model for long-horizon tasks.

However, some weaknesses exist: the approach relies on carefully tuned hyperparameters (k for epoch intervals and δ for Wasserstein distance); dependence on DAC specifically might limit generalizability to other AIL algorithms; the Wasserstein distance criterion adds computational overhead; the segment-based reward learning introduces another design choice that could impact performance; and the method assumes relatively monotonic improvement during AIL training, which might not hold in highly stochastic environments.

**Other Comments Or Suggestions:**

See above.

**Other Strengths And Weaknesses:**

While APEC represents an advancement in automated preference generation, several weaknesses deserve attention: the method introduces significant computational overhead through full AIL training and Wasserstein distance calculations without discussing efficiency implications; experiments are limited to continuous control tasks, raising questions about generalizability to discrete actions or goal-oriented problems; there's no concrete path toward offline settings despite acknowledging this limitation; the vulnerability to reward hacking remains unaddressed for complex environments; the approach depends on hyperparameters with limited sensitivity analysis or tuning guidance; and the ablation studies could be more comprehensive to clarify each component's contribution.

**Questions For Authors:**

1. Could the authors provide a comparative analysis of computational costs between APEC and baseline methods? This would help assess the practical implications of the added complexity from AIL training and Wasserstein distance calculations.

2. Have the authors explored APEC's performance on tasks beyond continuous control, such as discrete action spaces or goal-oriented environments? Understanding generalizability across diverse domains would strengthen the method's broader applicability.

3. How sensitive is APEC's performance to hyperparameter choices, particularly the minimum training epoch interval (k) and Wasserstein distance threshold (δ)? Additional analysis showing performance across different parameter settings would clarify robustness.

4. The theoretical analysis assumes monotonic improvement during AIL training. How does APEC perform when this assumption is violated due to training instabilities or high environment stochasticity?

**Relation To Broader Scientific Literature:**

APEC relates to the broader scientific literature in several key ways:

1. It extends reward learning from suboptimal demonstrations, building on work like AIRL while removing requirements for optimal demonstrations or direct human feedback.

2. It advances preference-based reward learning by addressing coverage limitations in methods like T-REX and D-REX, replacing noise injection with a principled approach using training progression.

3. It leverages theoretical properties of adversarial imitation learning (GAIL, DAC), particularly monotonic improvement during training.

**Theoretical Claims:**

I examined Proposition 3.1, which claims the value gap between expert and learned policies decreases with iterations. The theoretical claim effectively supports APEC's core design principle.

---

> ### Author Rebuttal · Authors · 2025-04-01
>
> Thank you for your valuable suggestions!
>
> **Q1**: Could the authors provide a comparative analysis of computational costs between APEC and baseline methods?
>
> **A1**: The table below shows the computation costs (training time) of all methods on the MuJoCo task. We can see that APEC and the other methods require similar training time, with the majority of time spent on policy learning using the learned reward. Additionally, we present the training time of each stage in the reward learning process. We find that calculating the W-distance incurs only a slight increase in training time.
>
>
> |  | SSRR | AIRL | DREX | LERP | APEC |
> | --- | --- | --- | --- | --- | --- |
> | reward learning | 4-6h | 2h | 2-4h | 2-4h | 4-6h |
> | policy learning | 24h | 24h | 24h | 24h | 24h |
>
> |  | SSRR | AIRL | DREX | LERP | APEC |
> | --- | --- | --- | --- | --- | --- |
> | AIL/BC training | 2h | 2h | 5min | 5min | 2h |
> | collect demos  | 2-4h | / | 2-4h | 2-4h | 2-4h |
> | compute distance | / | / | / | / | ~20min |
> | PBRL training | ~5min | / | ~5min | ~5min | ~5min |
> | total | 4-6h | 2h | 2-4h | 2-4h | 4-6h |
>
>
> **Q2**: Have the authors explored APEC's performance on tasks beyond continuous control, such as discrete action spaces or goal-oriented environments? Dependence on DAC specifically might limit generalizability to other AIL algorithms.
>
> **A2**: We conducted an additional experiment on the Atari game Pong, which features a discrete action space. Results show that APEC consistently outperforms both DREX and LERP in terms of task performance, demonstrating its broader applicability. Notably, for the Pong experiment, we used an alternative AIL method, IQLearn [1], to generate preference data, which further highlights APEC's generalizability across different AIL algorithms.
>
> | task performance | Demo Mean | DREX | LERP | APEC (ours) |
> | --- | --- | --- | --- | --- |
> | pong | 3.7 | -9.5 | -20.0 | **-0.44** |
>
> [1] IQ-Learn: Inverse soft-Q Learning for Imitation. NeurIPS’21.
>
>
> **Q3**: How sensitive is APEC's performance to hyperparameter choices?
>
> **A3**: We perform ablation studies on key hyperparameter choices, including the **training epoch interval**, **Wasserstein distance threshold**, and **segment length for reward learning**, across both feature-based HalfCheetah and pixel-based walker_run tasks. The "default" refers to the values used in our original experiments, and we show the overall task performance here, as suggested by the reviewer. The results show that APEC is not sensitive to these hyperparameters.
>
> - Epoch Interval
>
>     |  | 5 | 10 (default) | 50 |
>     | --- | --- | --- | --- |
>     | HalfCheetah | 11957 | **12232** | 10019 |
>     |  | 5 | 10 (default) | 20 |
>     | walker_run | **710** | 701 | 591 |
> - Distance Threshold
>
>     |  | 0.05 | 0.1 (default) | 0.2 |
>     | --- | --- | --- | --- |
>     | HalfCheetah | **13291** | 12232 | 8848 |
>     |  | 0.005 | 0.01 (default) | 0.02 |
>     | walker_run | 709 | 701 | **736** |
> - Segment Length
>
>     |  | 100 | 500 | 1000 (default) |
>     | --- | --- | --- | --- |
>     | HalfCheetah | 12658 | **13434** | 12232 |
>     |  | 5 | 10 (default) | 20 |
>     | walker_run | **712** | 701 | 696 |
>
>
> **Q4:** The method assumes relatively monotonic improvement during AIL training, which might not hold in highly stochastic environments.
>
> **A4**: We would like to clarify that the theoretical analysis (Proposition 3.1) does **not** assume monotonic improvement during AIL training. Instead, we indeed prove that the value gap bounds of policies learned during AIL process increase as the iteration index $k$ decreases. This suggests that the preference relationship between two policies holds if their corresponding iteration indexes differ sufficiently, which further inspired our algorithmic design.
>
> To handle with the case where the environment is highly stochastic, APEC leverages the Wasserstein distance metric to help choose the policy pair, which enhances the robustness of preference generation.
>
>
> **Q5**: Relatively high-quality demonstrations despite claims of challenging setup and exploration of APEC's performance with varying amounts of demonstration data.
>
> **A5**: To better understand how the quality and quantity of demonstrations impact overall task performance, we conduct experiments on four MuJoCo tasks under two conditions: lower quality demonstrations and more demonstrations. In each value pair “x/y”, x represents the performance of the demonstrations, while y is the performance of the policy trained with the reward model learned from APEC. The results demonstrate that APEC can still achieve a satisfying performance in tasks with low-quality and scarce demonstrations.
>
> | **Performance** | Ant | HalfCheetah | Hopper | Walker2d |
> | --- | --- | --- | --- | --- |
> | low 1 demo | 2413 / 1522 | 4232 / 7874 | 1325 / 3300 | 2205 / 2089 |
> | high 1 demo | 4105 / 5221 | 7313 / 12232 | 2275 / 3310 | 3706 / 5394 |
> | high 10 demo | 4179 / 4830 | 7385 / 14700 | 2388 / 3325 | 3728 / 5390 |

---

### Decision · Program_Chairs · 2025-05-01

**Decision:**

Accept (poster)

**Comment:**

This was one of the most controversial papers in my AC stack. At the start of the author-reviewer discussion period, reviewers were originally [2,2,3,4], but during reviewer discussion one reviewer increased their score to 3, leading to scores that are closer to a consensus accept: [2,3,3,4]. However, reviewer Gguy still believes the experimental scope is insufficient for acceptance to ICML.

**Summarizing reviewer comments:**

Strengths:
- “The paper's claim that later AIL policies are preferred over earlier ones is adequately supported through both empirical and theoretical evidence.” (Gguy)
- Theoretical results are correct (Gguy, YpdB)
- “The proposed methods are well-structured and appropriate for automated preference generation.” (Gguy)
- “the presented method is novel and interesting” (YpdB)
- “The experimental design is generally sound with key strengths: diverse benchmarks (MuJoCo and DMControl), clearly defined suboptimal demonstrations (50-80% of optimal), test dataset for generalization assessment, controlled sample sizes across methods, and appropriate evaluation metrics” (Gguy)
- “the selection of benchmarks and baselines (SSRR, LERO) appears to be appropriate” (aJ5F)
- “Overall methods and evaluations are plausible” (aJ5F)
- “The proposed study is simple yet effective. Despite its simplicity, it appears to be supported by appropriate theoretical proofs.” (aJ5F)
- “I think this paper has a broader impact on various research domains, including human preference learning and understanding (related to RLHF for LLMs), and areas like GAN training.” (LmzP)

Weaknesses:
- The practicality of the method, specifically the “significant computational overhead” and the fact that the method “relies on carefully tuned hyperparameters” (Gguy)
- Limited experiments
  - “restriction to continuous control tasks, relatively high-quality demonstrations despite claims of challenging setup, focus on correlation metrics rather than task performance, limited discussion of computational efficiency or training stability, and no exploration of APEC's performance with varying amounts of demonstration data.” (Gguy)
  - “only several simple benchmark problems are adopted to verify the effectiveness of the proposed approach” (YpdB)
  - requires more recent baselines (YpdB)

**Rebuttal:**

In the rebuttal, the authors provided a significant number of new experiments showing the computational time only adds 2 hours / 28, it works for discrete games (Atari), it's not sensitive to hyperparameters, the monotonicity assumption is not required, and they added an additional baseline. This appeared to me to address the main weaknesses identified during the review period.

**Decision:**

Given the move towards a consensus accept and the strength of the new experiments in the rebuttal, I am giving this paper a weak accept. The authors should include the new rebuttal results in the revised version of the paper.